# Coastal submesoscale processes and their effect on phytoplankton distribution in the SE Bay of Biscay

Xabier Davila[1], Anna Rubio[1], Luis Felipe Artigas[2], Ingrid Puillat[3], Ivan Manso-Narvarte[1], Pascal Lazure[3], and Ainhoa Caballero[1]

[1]AZTI Marine Research, Basque Research and Technology Alliance (BRTA), Herrera Kaia, Portualdea z/g, 20110 Pasaia, Spain
[2]Université du LIttoral Côte d'Opale, Université de Lille, CNRS UMR 8187 LOG, Wimereux, France.
[3]IFREMER/Dyneco/Physed, BP 70, 29280 Plouzané, France.

**Correspondence:** Xabier Davila (Xabier.Davila@uib.no)

**Abstract.**

Submesoscale dynamics have a determinant role in several ocean processes. Submesoscale processes transport momentum, heat, mass and particles, and can define niches where different phytoplankton species flourish and accumulate, not only by nutrient provisioning but also by modifying the water column structure or by active gathering through advection. However, this effect is not straightforward in coastal areas, where they coexist with different spatio-temporal scale oceanic processes. The present study brings into consideration the relevance of the dynamic variables in the study of phytoplankton distribution, from the analysis of in-situ and remote multidisciplinary data. In-situ data were obtained during the Etoile oceanographic cruise, which surveyed the CapBreton canyon area in the South-East of Bay of Biscay in early August 2017. The main objective of this cruise was to link the occurrence and distribution of phytoplankton spectral groups with submesoscale ocean processes. In-situ discrete hydrographic measurements and multi-spectral chlorophyll-a profiles were obtained in selected stations, while temperature, conductivity and in-vivo chlorophyll-a were also continuously recorded at the surface. On top of these data, remote sensing data available for this area, such as High Frequency radar and satellite data were also processed and analysed. From the joint analysis of these observations, we discuss about the relative importance and effects of several environmental factors on phytoplankton spectral groups distribution above and below the pycnocline, and at the DCM by performing a set of General Additive Models. Overall, salinity is the most important parameter modulating not only total chlorophyll-a but also the contribution of the two dominant spectral groups of phytoplankton, Brown and Green algae. However, at the Deep Chlorophyll Maximum, among the measured variables, vorticity is the main modulating environmental factor and explains most of the deviance for total chlorophyll-a and Brown algae distributions.

## 1 Introduction

The monitoring and characterization of submesoscale dynamics are determinant for the appropriate comprehension of marine ecosystems (Lévy et al., 2012). Submesoscale processes refer to those features that range on spatio-temporal timescales of $O(0.1 - 10)$ km and $O(1)$ day. The timescales at which these processes evolve make them uniquely important to the structure

and functioning of planktonic ecosystems (Lévy et al., 2012; Mahadevan, 2016). They interact with the ecosystem by either driving episodic nutrient pulses to the sunlit surface, by increasing the mean time that the photosynthetic organisms remain in the well-lit surface (Lévy et al., 2012), or by reducing and even suppressing the biological production (Gruber et al., 2011). The importance of this biophysical interaction extends beyond primary production. Since primary production absorb atmospheric $CO_2$, submesoscale processes might actively contribute to the carbon export and regulate the fate of particulate organic carbon (Mahadevan, 2014). The interaction between submesoscale processes and phytoplankton has also implications for regional biogeochemical budgets, plankton studies, fisheries, and management (Irigoien et al., 2007).

The interaction between ocean dynamics and phytoplankton covers a wide range of spatio-temporal scales, and these are inherent to the surveying strategy to be selected. D'Ovidio et al. (2010) linked the occurrence of different phytoplankton groups with the large-scale surface ocean dynamics, based on altimetry data. They defined the so-called fluid dynamical niches where the phytoplankton assemblages interact with distinct physiochemical environments. However, available satellite observations lack the spatio-temporal resolution to properly resolve the fast-evolving (sub)mesoscale coastal processes. In coastal regions, where oceanic currents meet the bathymetry, the connection between the (sub)mesoscale processes and phytoplankton becomes even more challenging, and therefore requires more demanding surveying methods that can cover a high range of spatio-temporal scales. Nowadays, autonomous gliders can typically cover 1 km horizontally in an hour, but even this can be too slow for synoptic measuremets of larger submesoscale features (O(10) km). An alternative is the use of ship-towed undulating devices, which allow sampling 10-20 times faster than a glider (Lévy et al., 2012). Contrarily, (sub)mesoscale to microscale vertical patterns of chlorophyll-a (chl-a) concentration have been studied widely by the use of in vivo fluorometric casts, allowing to identify the Deep Chlorophyll Maximum (DCM) (Cullen, 2015). Differences within the DCM in terms of concentration, biomass and diversity (Latasa et al., 2017) stress the importance of the environmental drivers involved, on which the occurrence of (sub)mesoscale processes play a critical role (Lévy et al., 2012).

This study focuses on the innermost South-Eastern region of the Bay of Biscay (SE-BoB), a semi-open bay limited by the Spanish coast in the southern part and the French coast in the eastern part. The BoB is an area of complex coastal hydrographic and hydrodynamic processes, mainly due to the intricate bathymetry, the seasonally modulated and episodically strong river runoff, the wind- and density-driven ocean circulation and their interplay. The circulation in the coastal SE BoB is controlled mainly by the prevailing winds, although the general pattern is characterised by a weak anticyclonic circulation in the central deeper region (Valencia et al., 2004; Pingree and Garcia-Soto, 2014). The wind pattern either reinforces or weakens the seasonal Iberian Poleward Current (IPC), which flows cyclonically over the slope in autum and winter (Rubio et al., 2013). The IPC is, by the interaction with the bathymetry, responsible for the generation of Slope Water Oceanic eDDIES (SWODDIES) (Caballero et al., 2016; Teles-Machado et al., 2016). Besides, the ocean surface layer in this region is subjected to the seasonal variations of the water runoff from the nearby rivers: Gironde, Loire and Adour (Reverdin et al., 2013). The river runoff significantly modifies the water mass adjacent to the shelf (Caballero et al., 2016) by creating turbid and dilution

plumes, which act as a nutrient source to the surface layers and sustain primary production in the region (Morozov et al., 2013).

These complex ocean dynamics can modulate phytoplankton occurrence in the BoB. The flow of the IPC generates a shelf-break convergent front that separates the advected high-salinity and warm waters from the cold fresher coastal waters. The vertical mixing associated with this frontal system had a substantial influence on the whole plankton community (Fernández et al., 1993). Caballero et al. (2016) reported a Deep Chlorophyll Maximum (DCM) in the centre of a SWODDY resulting from the vertical velocities and eddy-wind induced Ekman pumping in the centre of the anticyclone. More recently, Muñiz et al. (2019) described the phytoplankton annual cycle on the SE-BoB and reported that temperature and nutrients explained most of the of the variability of chlorophyll concentration. Nevertheless, to our knowledge, none of these studies have focused on the relative importance of different hydrographic and hydrodynamics forcing mechanisms, especially while focusing on the submesoscale dynamics.

In order to the shed light on the coastal submesoscale dynamics and its effects on chlorophyll-a (chl-a) and phytoplankton spectral groups distribution, the Etoile oceanographic cruise surveyed the CapBretron Canyon area on early August 2017. This cruise was one of the research actions in the framework of the European H2020 Joint European Research Infrastructure for Coastal Observatory - Novel European eXpertise for coastal observaTories (JERICO-NEXT) project. The regional coastal observatories (EuskOOS) are also embedded in the JERICO Research Infrastructure and provide the operational HF radar data complementing the Etoile in-situ measurements. Both JERICO-NEXT (2014 - 2019), its predecessor JERICO (2007–2013) and the ongoing JERICO-S3 (2020-2024) aim to the consolidation of a pan-European coastal observatory infrastructure, for a better understanding of the functioning of coastal marine systems and a better assessment of their changes. In this study, we first describe the submesoscale processes that are present in the study area based on the joint analysis of a wide range of multiplatform spatio-temporal data, from the remote sensing data available for this area to in-situ measurements. Secondly, we investigate the link between the observed submesoscale structures and the distribution of the two dominant spectral groups of phytoplankton by performing a set of General Additive Models. Thereby, assessing the relative importance and effects of several environmental factors, including the submesoscale processes, on phytoplankton distribution above and below the pycnocline, and at the DCM.

## 2 Material and Methods

### 2.1 In-situ data from the Etoile cruise

In the framework of the European H2020 Joint European Research Infrastructure for Coastal Observatory - Novel European eXpertise for coastal observaTories (JERICO-NEXT) project, the "Côtes de la Manche" research vessel (CNRS-INSU), surveyed the area of CapBreton canyon from August $2^{nd}$ to $4^{th}$ 2017 during the Leg 2.2 of the Etoile oceanographic cruise (P.I. Pascal Lazure, IFREMER, DOI: 10.17600/17010800), aiming to undercover the mesoscale dynamics. The cruise consisted in six transects covering the continental shelf and slope, as well as the axis of the canyon, as shown in Figure 1. During east-west

transects (T1, T3 and T5) a CTD (Sea-Bird) was deployed every 7 km, while during west-east transects (T2, T4 and T6) a Moving Vessel Profiler (MVP200 operated by Genavir) was towed and the profiles were averaged every 5 km. As the spatio-temporal distribution of the observations is important, the use of the Moving Vessel Profiler (MVP) allowed a more extensive and quicker sampling suitable for small, rapidly evolving structures.

During the cruise, chlorophyll-a (chl-a) was estimated by a FluoroProbe (Bbe Moldakenke) multi-spectral fluorometer, which measures chl-a and accessory pigments using LEDs with different wavebands. Therefore, it is possible to distinguish between four algal pigmentary groups: "Blue algae" (e.g. Cyanobacteria), "Green algae" (e.g.Chrolorophytes, Chrysophytes), "Brown algae" (e.g. Diatoms, Dinoflagellates) and "mixed red group" (e.g. Cyanobacteria, Cryptophytes). It estimates chl-a equivalent concentrations for these four groups and total chl-a following the algorithms of Beutler et al. (2002) as explained in MacIntyre et al. (2010) and a manufacturer's calibration, and also provides an estimation of the concentration of chromophoric dissolved organic matter (CDOM or yellow substances). Up to 80 m depth in-vivo fluorescence profiles were obtained. Unfortunately, this data could just be gathered in the T3 and T5 transects due to connection issues with the instrument during T1. During the whole cruise salinity and in-vivo fluorescence were continuously measured on surface (3.5 m deep) by a thermosalinograph and a second automated multi-spectral fluorometer, respectively. This continuous record at surface provided valuable qualitative information, since the resolution of satellite data is too low at the submesoscale.

## 2.2 Complementary operational remote sensing data

In addition to the in-situ data, remote sensing operational data was used to complete the picture obtained during Etoile. Ocean surface currents measurements were obtained by two long-range HF radar antenna located at Matxitxako and Higer Capes. The antennas are owned by the Directorate of Emergency Attention and Meteorology of the Basque Security Department, and are part of EuskOOS network (euskoos.eus/en). They emit at a central frequency of 4.463 MHz and a 30-kHz bandwidth, and provide hourly horizontal currents maps (corresponding to vertically integrated horizontal velocities in the first 3 m of the water column) (Rubio et al., 2013). The receiving signal, an averaged Doppler backscatter spectrum, allows to estimate surface currents over wide areas (reaching distances over 100 km from the coast) with high spatial (1-5 km) and temporal ($\leq$1 h) resolution (Figure 1B). To obtain the surface velocity data we followed the methodology of Rubio et al. (2013). Velocity data is processed from the spectra of the received echoes every 20 minutes using the MUSIC (MUltiple SIgnal Classification) algorithm. Then, a centred 3h running mean average was applied to the resulting radial velocity fields as part of the pre-processing previous to the computation of total currents. The current velocity data were quality controlled using procedures based on velocity and variance thresholds, signal-to-noise ratios, and radial total coverage, following standard recommendations (Mantovani et al., 2020). The performance of this system and its potential for the study of ocean processes and transport patterns have already been demonstrated by previous works (e.g. Solabarrieta et al. (2014, 2016); Rubio et al. (2011, 2018)).

In order to visualize representative velocity fields, we applied a 10th order digital Butterworth low-pass filter (Emery and Thomson, 2001) to both velocity components at each node (filtering out T<48 h). Therefore, HF processes such as inertial currents or tides were removed, as these are irrelevant for this study and would have eclipsed the mesoscale geostrophic and synoptic wind-induced current patterns. A Lagrangian Particle-Tracking Model (LPTM) was applied to HF radar data to simulate trajectories and analyse surface ocean transport patterns around the dates of the Etoile survey. Particles released within the HF radar coverage area were advected using a $4^{th}$ order Runge–Kutta scheme (Benson, 1992). In this case, the particles are advected using the 2D hourly current fields given by the HF radar from July $26^{th}$ to August $11^{th}$. To describe (sub)mesoscale patterns, Lagrangian Residual Currents (LRC) were calculated following a methodology similar to that described in (Muller et al., 2009), using an integration time of 3 days.

Furthermore, satellite data prior to and after the cruise was also analysed. SST and Chlorophyll-a data were retrieved from the Visible and Infrared Imager/Radiometer Suite (VIIRS) sensor and water turbidity, from MODIS. In addition to these datasets, hourly wind information of July and August was collected by the mooring buoy of Bilbao owned by Puertos del Estado (available in www.puertos.es). Although its location is not exactly in our study area (Figure 1B) it is considered close enough for a general description of the wind regime in the bay.

### 2.3 Computation of vorticity and vertical velocities

From hydrographic data alone, geostrophic circulation can be diagnosed, inferring various key dynamical variables such as geostrophic relative vorticity (hereinafter referred to just as vorticity) or the vertical velocity from a 3D snapshot of the density field. To compute vertical velocities, we assume quasi-geostrophic dynamics and a synoptic or steady state, where the Rossby number is small (Ro = U/f L «1, where U is the characteristic velocity, L is length scale and f the Coriolis parameter) and submesoscale features remain constant during the sampling (Gomis et al., 2001). To reduce the computational effort during the analysis of the data, the MVP transects were averaged every 5 km, considering it enough resolution for resolving submesoscale structures, following the methodology in (Gomis et al., 2001). An interpolation of the data allows deriving key dynamical variables, such as the geostrophic relative vorticity and vertical velocities. This was accomplished by merging the CTD and averaged MVP profiles after verifying that no significant bias was present between the measurements of these two instruments. Once having verified that data can be merged, the Optimal Statistical Interpolation (OSI) was performed by the 'DAToBJETIVO' software package developed by Gomis and Ruiz (2003), for the objective spatial analysis and the diagnosis of oceanographic variables.

For the interpolation in the sampling area, an 11 x 33 output grid was used with a 0.031° x 0.033° resolution (Figure 1A), pursuing a compromise between providing a good representation of the scales that can be resolved by the sampling and minimizing the effect of the observational error. A Gaussian function for the correlation model between observations (assuming 2D isotropy) was set up, with a correlation length scale of 15 km. The noise-to-signal (NTS) variance ratio used for the analysis

of temperature, salinity and dynamic height were: 0.01, 0.005, and 0.0027, respectively. This ratio was defined as the variance of the observational error divided by the variance of the interpolated field (the latter referring to the deviations between observations and the mean field). This parameter allows the inclusion in the analysis an estimation of the observational error and adjustments of the weight of the observations on the analysis (the larger the NTS parameter, the smaller the influence of the observation). Then, after the interpolation, all fields were spatially smoothed, with an additional low-pass filter with a cut-off length scale of 10 km to avoid aliasing errors due to unresolved structures. This resulted in a coarse grid that allowed the appropriate representation of the subsequent spatial derivatives of the analysed field Gomis et al. (2001). In the vertical, 98 equally-spaced levels were considered, from 4 to 200 m (every 2 m). To analyse and correlate the explanatory and the response variables, the same interpolation was performed for the fluorescence data.

## 2.4   Statistical Analysis

The presence of a seasonal pycnocline determines the occurrence of two dynamically different sections that need to be analysed separately to constrain the different dynamical environments. Therefore the dataset was divided between three different dynamics areas of the section "Above the pycnocline", "Below the pycnocline" and at the DCM, according to the dynamical environments.

The environmental variables were compared to fluorescence data, after being interpolated to a common grid. We assessed the relative importance of different environmental factors involved in the phytoplankton distribution by developing a General Additive Model (GAM) statistical model (Hastie and Tibshirani, 1990). GAMs offer the possibility of identifying non-linear relationship between variables by the inclusion of a smoothing function that has no specific shape. Since the relationship among variables along the entire water column might mask each other, three GAMs were implemented for the different dynamical environments in the water column, the section "Above the pycnocline", "Below the pycnocline" and at the Deep Chlorophyll Maximum (DCM) (Equation 1):

$$[Chl-a]_z = a + g_1[Sal_z] + g_2[Temp_z] + g_3[Vor_z] + g_4[V.Vel_z] \tag{1}$$

Where $a$ is an intercept, $z$ is the location in the water column (above or below the pycnocline or at the DCM) and the $g$s are nonparametric smooth functions describing the effect of environment on chl-a concentrations. Sal, Temp, Vor and V.Vel correspond to the environmental variables determined in this study, salinity, temperature, vorticity (cyclonic/anticyclonic) and vertical velocities (upwelling/downwelling), respectively.

In order to account for co-linearity problems, we calculated pairwise Spearman correlation coefficients (r) between variables. The only pair of variables correlated were salinity and temperature for the section below the pycnocline (r = -0.77, *p-value*<0.05) related to the depth dependency of both variables. The model selection was based on the analysis performed

by Llope et al. (2009) where a stepwise approached was implemented by removing covariates and minimizing the generalized cross-validation (GCM) criterion of the model (Wood). The GCV criterion is a measure of the out-of-sample predictive performance of the model and is related to the Akaike's Information Criterion (AIC) (Wood, 2006). Similarly, by deleting one variable at a time we can quantify the penalty on the explained deviance of the phytoplankton distribution Llope et al. (2009). In total 12 GAMs were carried out from the combinations of Total chl-a, Green chl-a, Brown chl-a and Brown:Green ratio (B:G) among the three selected subset on the water column (Table 1). All the variables have a significant impact on the total and group chl-a distribution except the vorticity for the green algae chl-a at the section above the pycnocline. If vorticity is removed, the model slightly improves (GCV decreases from 0.013 to 0.0125; Table 2). Yet, we decided to keep it in the model, having in mind that offers no improvement. The rest of the variables, even if they explain a small part of the deviance, they significantly improve the model. The GAMs were carried out by the R (version 3.63, R Core Team (2020)) and the package used was mgcv (version 1.8.33) (Wood, 2011).

## 3 Results

### 3.1 Mapping coastal mesoscale hydrography and currents

The combined use of wind data and satellite imagery together with the HF radar provide a context of hydrographical and dynamical regime around the dates of the Etoile cruise. Figure 2 shows the Progressive Vector Diagram (PVD) of the wind conditions. From July $21^{st}$ to the $28^{th}$, the predominant wind had a marked north-westerly component with relatively high intensity. Afterwards, it decreased in intensity, shifted and started blowing from the north-east. On the August $7^{th}$ the wind had again a north-west component for few days. Then, the wind conditions during the whole cruise remained variable in direction but low in intensity. Figure 3 shows the satellite SST, chl-a and turbidity fields, the latter allow us to locate the river plumes of the Adour and the Bidasoa rivers. In addition, the LRC fields derived from the HF radar, and superimposed to the previous fields, give a high-resolution image of the surface transport for the previous 3 days to the survey, encompassing two periods, July 26-29$^{th}$ and July $30^{th}$ to August $2^{nd}$. The LRC provide an essential insight about the time-evolving surface circulation and position of the river plumes. The surface current pattern also shifts during the cruise. Prior to Etoile (Figure 3 – left column), during north-westerly winds the circulation is somewhat chaotic. This circulation regime shows two cyclonic eddies, located in our sampling area (C17W at 43.6$^o$N and 2$^o$W and C17E at 43.7$^o$ and 1.7$^o$W). During the second period (Figure 3 – right column), the shift to north-easterly winds generates a remarkable transition to westward currents. At this moment, the eddies are not visible by the HF radar. And instead, in their position we observe a meandering pattern that affects the distribution of the SST, the position of the river plumes and their associated chl-a distribution. In addition, on August $2^{nd}$ a sharp change in SST is observable close to the French inner shelf, which is linked with the upwelling generated by the north-easterly winds.

During the Etoile cruise (August $2^{nd}$ to $4^{th}$), the first meters of the water column are characterized by a high spatial variability (Figure 4). Although the river plume is not visible anymore in the salinity fields at 14 m, a layer of relatively fresh water is located in the inner continental shelf (1.6 – 1.7$^o$W). This low salinity front extends over 20 km horizontally and 18 m vertically

(Figure 5A) if we consider the boundary at a salinity of 35.1 (Puillat et al., 2006). At 60 m depth (Figure 4, a second salinity front is observed, this time coinciding with the shelf break, located approximately along the - 250 m isobath with a vertical extension between 50 and 120 m (Figure 5). Here the fresher waters ($<$35.5) occupy the totality of the water column on the shelf, in comparison with the oceanic water at the slope ($>$35.6). The salinity range in the shelf break front is much smaller than in the surface front.

The sampling was carried after the wind shifted to a north-easterly component. The presence of the cyclones depicted in Figure 3 is also evidenced at deeper layers. While it is not appreciated in the temperature fields, their existence is clearly rep-resented by vorticity and geostrophic velocities and heir position coincide with the ones observed by the HF radar during the period $26^{th}$-$29^{th}$ of July, just before the cruise. The disappearance of the C17W and C17E in the LRC fields during the cruise period coincides with a change in the wind pattern, which resuts in a surface wind-driven flow that masks the geostrophic circulation at surface. Few days after, once the wind changes back to a north-west component, C17W is observable again in the HF radar (See Supplementary Material Figure A1), suggesting a persistent nature. Noteworthy, the vorticity fields also show an anticyclone (A17) at the NW part of the domain (centred at $43.80^{o}$N $2.25^{o}$W), although this is not observed in the HF radar fields. At greater depth the cyclones are still noticeable, although they become progressively weaker. In addition to the anti-cyclone A17, a region of anticyclonic vorticity is well defined in the frontal area between the cyclones. At 60 m the cyclonic eddies present a negative temperature anomaly and relative higher salinity values. The eddy A17 is associated to a positive temperature anomaly and higher salinity. Associated to the frontal areas in the two dipoles (A17-C17W and C17W-C17E) we observe main upwelling areas (positive vertical velocities) which maxima have a relatively constant position throughout the water column.

From the cross-section at $43.77^{o}$N, we can observe the vertical extension of both the low salinity surface front and the shelf break salinity front (Figure 5A). The surface salinity front has a vertical extension of $\sim$20 m, while the location of the shelf break front ranges from $\sim$50 to 110 m. The uplift and depression of the isopycnal lines (black contours) is coherent with the presence of submesoscale structures of different polarity, mostly following the temperature distribution. These two variables contribute to the water density and the position of the seasonal pycnocline at $\sim$25 m, primarily conditioned by the warming of surface waters in summer. From the vorticity field and the geostrophic meridional velocities (Figure 5D), it is noticed that the position of the anticyclonic frontal area between C17W and C17E coincides with the shelf break ($1.9^{o}$W) and its strength decreases with depth from a maximum at  25 m. The onshore area is dominated by a southward flow while the offshore area is dominated by a northward. As in Figure 4, the highest vertical velocities are located in the eddies' periphery, where the largest vorticity gradients are located.

## 3.2 Chlorophyll -a and spectral groups distribution

Chl-a and salinity data collected at surface by the continuous recording system provides a synoptic distribution of the phytoplankton during the sampling period (August $2^{nd}$ - $4^{th}$ 2017). Figure 6 illustrates how chl-a distribution is spatially dependent on the position of the river plume at 3.5 m depth. The maximum fluorescence is observed around the salinity minimum, decreasing to the NW (and with the depth) in accordance with the increase of salinity.

The cross-section along the hydrographic and hydrodynamic data at 43.77°N, provides an image of the spatial distribution of Total chl-a and spectral groups (Figure 7). The distribution is complex, two Deep Chlorophyll Maximum (DCM) are observed, one at the inner shelf at ~30-50 m, and the second located at the shelf edge at 50-65 m, below the pycnocline. At the same time, the shallower DCM seems to be split into two cores, although its morphology is hard to asses due to the limited spatial coverage of the sampling. However, this patchy distribution correlates at some extent with the areas of minimum vorticity. The deepest DCM, at the shelf break, is located at anticyclonic frontal area between C17W and C17E. Regarding the composition of the DCMs, a major fraction of the Total chl-a results from the brown algae, the dominant spectral group. The maximum is again centred in the anticyclonic frontal area, particularly between C17W and C17E. Green algae, however, follow a different pattern and are distributed slightly deeper, following the salinity contours at waters saltier than 35.55. The ratio between brown algae and green algae (B:G), logarithmically transformed, provides an even clearer image of how the different spectral groups are distributed. There is a sharp transition between the brown algae dominating an area around the anticycloninc frontal area and the green algae dominating the area below the 35.55 isohaline. A cross-section at the 43.70 $^o$N (See Suplementary Material Figure A2), out of the core of the anticyclonic frontal area, reveals that this pattern is not ubiquitous. Here, there is not a clear dichotomy among the groups nor a deeper maximum of green algae.

## 3.3 Exploring bio-physical interactions

The set of GAMs shown in Figure 8 corresponds to the section "Above the Pycnocline", showing the partial additive effect that the term on the x-axis has on the chl-a. Overall, all the models perform well, explaining in all cases more than 40% of the deviance (Table 1). As expected, and in agreement with Figure 6, low salinity values are correlated with Total chl-a concentration, as well as warmer waters (Figure 8a and 8b). Salinity and temperature explain most of the deviance of the model, and their contribution to the explain the 13.10% and 9.8% of the deviance, respectively (Table 2). Oppositely, the contribution of vorticity (Figure 8c) and vertical velocities (Figure 8d) to the distribution of Total chl-a is very small. The straight vertical distribution around 0 vertical velocities in Figure 8d (as well as in Figure 8h, 8m and 8p) is an artefact of the surface boundary condition necessary to perform the calculations, where velocities are assumed to be null. The distribution of Brown algae chl-a differs from the Total chl-a. Fresher waters affect negatively this group, while the effect is positive at salinity values around 35.1 (Figure 8e). The effect of temperature is positive at colder water temperature values and negative at warmer values (Figure 8f). There is a positive response of Brown algae chl-a to positive values of vorticity that represent cyclonic circulation. Still,

salinity explains most of the deviance of the model, %23.3. However, and as a difference to the Total chl-a, vorticity explains more of the deviance than temperature, 12.4% and %7.5, respectively. The response of the Green algae chl-a almost mimics the one of the Total chl-a (Figure 8i-m), with the difference that the relation with vorticity is non-significant (*p-value*>0.05). The main environmental factor shaping the Green algae chl-a distribution is the temperature, explaining %10.40 of the deviance. Even if salinity alone explains 12.70%, its removal from the model has no penalty in its explained deviance, likely owing to the temperature capturing most of the variability. Besides, the B:G ratio shows a response mostly to temperature and salinity (Figure 8n and 8l), and the positive effects peak at the values where the Brown algae and Green algae chl-a distributions are out of phase, where fresher and warmer waters have a strong differential effect, correlated with the increase of Green chl-a versus Brown chl-a. The contributions of both salinity and temperature to the deviance are similar, 15.6% and 15.8%, respectively. In all the four GAMs for this section, vertical velocities explain only ≤2% of the deviance.

The set of GAMs computed for the section "Below the pycnocline" explains a larger percentage of the deviance and generally performed better than for the section above, except for the Green algae chl-a model which explains almost 10% less of the deviance. The response of the different chl-a distributions to the environmental factors is slightly different than "above the pycnocline". The negative trend of lower chl-a values with increasing salinity observed in Figure 8a continues until it reaches values of ∼35.5 in Figure 9a. Temperature exerts the greatest impact, being mostly positive at ∼14 °C (Figure 9b) and capturing 34% of the explained deviance, while the rest of the environmental factors explain about 2% or less (Table 2). Although the explained deviance by vorticity is small, there is a clear trend (Figure 9c). Positive effects in chl-a distributions are correlated with negative values of vorticity (anticyclonic circulation) and negative impact with positive vorticity values (cyclonic circulation). Similarly, the responses of Brown algae chl-a mimic the ones of the Total chl-a. Although temperature also explains most of the deviance, 27.30%, in contrast with the Total chl-a, 8.80% of the deviance is explained by salinity. For Green algae, chl-a distribution is positively affected by salinity, showing a linear relationship (Figure 9i). The effect of temperature (Figure 9j) is most positive at slightly colder waters than for Total chl-a and Brown algae chl-a. Temperature explains most of the deviance for Green algae chl-a as well, 19%. The effect of vorticity (Figure 9k) and vertical velocities (Figure 9m) and does not offer a clear response, reflected in their explained deviance, 2.10% and 1.3% respectively. Regarding the B:G ratio, Figure 9n shows a trend from higher concentration of Brown chl-a at low salinity waters to higher Green chl-a concentrations at saltier waters. The effect of temperature is mostly cancelled out in Figure 9l and shows a negative effect of low temperatures in the B:G ratio, correlated with higher Green chl-a concentration compared to Brown chl-a. In this case, salinity is the main factor modulating the differences between the Brown and Green chl-a, explaining 11.80% of the deviance, while temperature only explains 4.50%. Although there is also a trend in the response to vorticity, it only explains only 4.20% of the deviance.

Regarding the DCM, there is a larger difference between how the different models perform. The GAM for the Total chl-a only explains the 17.3% of the deviance. However, when this is separated between the spectral groups, the models improve substantially. The GAM for Brown chl-a explains 37.7% of the deviance and the one for Green chl-a 56.9%. The model for the B:G ratio explains even a higher percentage of the deviance, 64.5%. Saltier waters have a positive effect on the chl-a concen-

tration (Figure 10a), although the data at these values is very sparse. Temperature shows a similar peak (Figure 10b) to the one covering the whole section below the pycnocline (Figure 9b). The effect of vorticity on Total chl-a also (Figure 10c) shows a similar shape as in Figure 9c, and the same goes for vertical velocities (Figures 9d and 10d). However, the relative importance of the variables is different, vorticity explains the 9.97% and is depicted as the main modulating environmental factor, although very close to the 8.98% explained by salinity. The differences regarding the relative importance of the variables with other sections in the water column are reinforced for the Brown algae chl-a model. The effect of salinity and temperature in the DCM for Brown chl-a is almost flat (Figure 10e-f), and the explained deviance is very small, 1.6% and 0.7% respectively. Vorticity, on the other hand, is responsible of the 19.30% of the deviance and is strengthened as the main modulating factor in the DCM for Brown chl-a distribution. Although vertical velocities only explain 4.40% of the deviance, their impact is stronger than the one for salinity and temperature. The modulating factors for the Green chl-a differ substantially. Salinity has a great impact where higher chl-a concentration is positively affected by saltier waters (Figure 10i) and explains 25.40% of the deviance. The temperature effect resembles the ones for the below the pycnocline section (Figure 9j) with the strongest effect at ~14°C. The effect of vorticity shows an opposite response for Green chl-a (Figure 10k) than for Brown chl-a (Figure 10g). Nevertheless, the contribution of these two variables on the deviance explained by the model is small (≤6%). In fact, vertical velocities exert a higher impact on the Green chl-a distribution, where positive values (upwelling) impact negatively the chl-a concentration, explaining 10.5% of the deviance. The differential effect of the environmental factors on these two spectral phytoplankton groups are recorded on the response of the B:G ratio. Salinity stands out as the main modulating factor, explaining 20% of the deviance. However, the effect of vertical velocities is also considerable, 14%, as well as the one of vorticity, 9.30%.

## 4   Discussion

Prior to the Etoile oceanographic cruise, two cyclones (C17W and C17E) were observed by the HF radar, which disappeared from the surface signal when the wind pattern changed by the time the cruise took place. However, their signature remained at the subsurface and could be diagnosed from the hydrographic measurements obtained during the Etoile cruise. Thus, the geostrophic circulation indicated the presence of the dipole structure (C17W and C17E), with a region of anticyclonic circulation in between, and an additional anticyclone (A17). Further, a salinity front at the near-surface (<14 m) and at the subsurface (>50 m) were observed. From the chl-a observations, a DCM was located below the pycnocline at ~60 m, while the distribution of the two dominant spectral groups of algae, Brown and Green algae, was depicted. The relative importance of the environmental factors modulating the chl-a distribution was assessed with GAMs. The GAMs showed not only that these environmental factors affect the Brown and Green algae differently, but also that their relative importance changes throughout the water column. While salinity and temperature explain most of the deviance above and below the pycnocline of both Brown and Green chl-a, vorticity captures most of the deviance in the DCM for Brown algae.

## 4.1 Physical Environment

The hydrographic and hydrodynamic regimes at the SE-BoB during the Etoile cruise, despite being spatio-temporally highly complex, were not exceptional and similar conditions have been already recorded. The surface salinity front we encountered onshore was observed on early May 2009 by Reverdin et al. (2013). They described a fresher (34-35) and deeper (∼30 m) freshwater layer originated due to winter and spring river runoff and which signal weakens towards August by increasing salinity to ∼35, as a result of vertical mixing and offshore advection by Ekman transport. This shelf break front seems to correspond

to a persistent feature in the study area, which is originated due to the differences between the waters over the French shelf and the Landes Plateau and those located over the Spanish shelf and slope (Valencia et al., 2004).

Furthermore, the dipole-type structures have also been observed before in the BoB, yet in a larger scale (Pingree and Garcia-Soto, 2014; Solabarrieta et al., 2014; Caballero et al., 2016; Rubio et al., 2018). Both the location of the vertical velocities

at the periphery of the structures and the magnitude (O(1-10) m/day) are consistent with already reported results (Mahadevan et al., 2008; Lévy et al., 2012; Caballero et al., 2016). While the cyclones were detected by the HF radar before the cruise, these were vanished when the circulation changes due to the change in the wind regime. Their intermittent signature in the HF radar surface fields is explained by the interaction of the geostrophic and wind-induced flow. A similar situation was described using an analytical model in the Florida current by Yonggang et al. (2015), where a surface meandering flow was observed as a result

of the overlap between a coastal jet and an eddy dipole field. This is coherent with our observations, i.e. under predominant NE winds the wind-driven circulation over the eddy field results in a meandering structure. Indeed, as the wind weakens the cyclones signature is again observed in the HF radar fields, highlighting the importance of using a wide range of multiplatform spatio-temporal data for a better characterization of the coastal hydrodynamics.

## 4.2 Environmental Drivers

The described hydrographic and hydrodynamic conditions affect significantly the Total chl-a and phytoplankton spectral groups distribution, although the interplay between phytoplankton and environmental drivers seems to vary significantly depending on the location in the water column. In addition, coastal chl-a is highly dependent on the seasonality of riverine nutrient inputs in the BoB (Guillaud et al., 2008; Borja et al., 2016; Muñiz et al., 2019). From satellite imagery and continuously recorded

surface salinity and chlorophyll-a data (Figure 3 and 6), it is evident that the Adour and Bidasoa plumes are associated with the highest chl-a concentrations in the sea surface. Simultaneously, the location of the Adour and Bidasoa plumes is dependant of the wind conditions, which controls the non-geostrophic surface circulation as shown by the LRC derived from the HF radar. Our results agree with the observed general pattern in which westerly winds push the river plume towards the coast, while easterly winds promote an offshore expansion (Petus et al., 2014). Thereby, the surface-most chl-a pattern is eventually dependant

on the winds that modulate the position of the river plume. At subsurface, the occurrence of the DCM agrees with previously described phytoplankton distribution. Muñiz et al. (2019) described a DCM below 30 m in summer at the same sector on the

BoB. Caballero et al. (2016) also reported a summer DCM at around 40 m (below the thermocline) at the periphery of two cyclones.

Below the surface-most layer, the two major phytoplankton spectral groups respond differently to the different environmental factors. And in addition, the relative importance that each of the specific factors has for the chl-a distribution changes depending on the depth. For the statistical analysis, the dataset was divided into three different dynamics areas of the water column: "Above the pycnocline", "Below the pycnocline" and at the DCM. Above the pycnocline, non-geostrophic processes related to wind-driven currents (e.g. offshore advection of coastal waters during upwelling-favourable winds) have an impor-
tant role in the chl-a distribution changes, showing decreasing intensity with depth. On the contrary, below pycnocline, we could expect geostrophic currents progressively become the main driver for particle advection. These two sections are also different regarding the nutrient supply. Typically, waters above the mixed layer are depleted in nutrients, whereas below, the phytoplankton would benefit from the nutrient supply by ocean deep waters in combination with maximum light penetration in summer (Cullen, 2015). This can also lead to different phytoplankton communities with different nutrient requirements.

    Above the pycnocline, on the one hand, most of the deviance of Total chl-a and Brown algae chl-a is explained by the salinity; on the other hand, the environmental variable that explains most of the Green algae chl-a deviance is the temperature. Noteworthy, the highest chl-a concentrations occur mainly at surface fresher and warmer waters. These results suggest that the Green algae is likely associated to the presence of nutrients on river plumes (fresher and warmer waters) from Adour
and Bidasoa. However, the Brown algae seemed to be unaffected by the river plume, at least directly, since they display high chl-a concentration at deeper, colder and slightly saltier waters. The causative link between the environmental variables and the Brown chl-a distribution is harder to draw. Yet, salinity is the main modulating factor and might suggest an indirect link with nutrient provisioning by river runoff. In any case, the sharp change in the ratio by both salinity and temperature indicates a transition to a different environmental niche. Below the pycnocline, temperature is the variable that explains most of the
deviance. However, this could be the result of the positioning of the DCM at a specific depth and the large vertical gradient of temperature in the water column, where there would be a good compromise between light and nutrient availability (not measured during this study) for phytoplankton growth (Cullen, 2015). In fact, for the B:G ratio, this effect cancels out and salinity is the most important environmental factor. Similarly, it is observed that Green algae are more concentrated at deeper and colder waters. Regarding salinity, the responses of both Total chl-a and Brown chl-a are an extension to the responses in
the section above the pycnocline. The negative trend from 35.1 in Figure 8e is still present, but the effect of salinity is positive again at values equal and higher than 35.6, probably due to higher nutrient levels in deeper waters. Overall, when integrating to the entire water column, even though the responses differ in the different sections, salinity is the most important environmental factor regarding the Total chl-a distribution and the relative occurrence of Brown and Green algae. We attribute this effect to salinity and its relation to nutrient content at the surface fresher and at the deeper saltier waters (Muñiz et al., 2019).

While the GAMs for Total chl-a both above and below the pycnocline have performed satisfactorily, at the DCM, the GAM covering the Total chl-a performs substantially worse and only explains 17.3% of the deviance. Once the Total chl-a has been divided into Brown and Green algae chl-a (the two other categories representing less than 1% of Total chl-a), each of the respective GAMs improve. Only then, we are able to identify the role of salinity as the main modulating factor for Green algae chl-a and the B:G ratio, also visible at Figure 7c where chl-a concentration follows the 35.55 contour. Vorticity is the factor that explains most of the deviance in Total chl-a and Brown algae chl-a concentrations. The more negative (positive) the vorticity, the more anticyclonic (cyclonic) is the circulation and the more positive (negative) is the effect on Brown algae chl-a concentrations. In anticyclones, due to Ekman transport, a small part of the flow targets the core leading to an accumulation of phytoplankton at their center (Mahadevan et al., 2008). Oppositely, Ekman transport results in an outward transport in cyclones. Therefore, the cyclones C17W and C17E would have advected the Brown algae and expelled them from the core. These were then subsequently trapped in the anticyclonic circulation located between the cyclones. A similar pattern is described by Caballero et al. (2016), where the highest chl-a concentrations were located at the periphery of the cyclones. The effect of this advection by submesoscale processes is such, that the distribution of Brown algae at the DCM cannot be statistically explained without the addition of vorticity to the GAM.

However, the distribution of Green algae chl-a is not affected by vorticity, and the environmental factor that exerts most of the difference between the two spectral groups is salinity. From our observation we cannot explain the occurrence of a single spectral group in the core of the anticyclonic circulation. Latasa et al. (2017) demonstrated that, during the summer stratification in the Iberian Shelf and Margin, the DCM are composed of different types of phytoplankton, each of them adapted to the different existing micro-environments. However, the phytoplankton landscape organized in submesoscale patches are often dominated by a single species D'Ovidio et al. (2010). This structuring of the phytoplankton community is a direct effect of the horizontal stirring, which can create intense patchiness in species (Lévy et al., 2012). We believe that the observed submesoscale processes during the Etoile cruise would have perturbed an already existing horizontal layer of DCM, not enhancing primary production (not measured during our study) by themselves, but rather isolating, advecting and gathering the phytoplankton in the region of anticyclonic circulation.

## 4.3  Limitations of the study

It is worth remarking the main limitations encountered during this study, especially focusing on the Etoile cruise. The area covered by the sampling was insufficient for completely resolve some of the observed structures. Similarly, having just a synoptic image of the processes and lacking temporal information (despite operational and remote sensing data) makes challenging to derive a cause-consequence relation, especially regarding the evolution of the system. Although we use of chl-a as a proxy for phytoplankton biomass concentration, we note that photo-acclimation of pigment content (Cullen, 2015), which, together with variable fluorescence to chlorophyll ratios (Estrada et al., 1996; Kruskopf and Flynn, 2006; Houliez et al., 2012), could lead to

elevated chl-a concentration relative to phytoplankton biomass at depth.

In addition, no further phytoplankton classification was carried out which might have helped defining specific environmental niches (D'Ovidio et al., 2010; Latasa et al., 2017) and correlating spectral groups to pigmentary groups and/or taxa. The latter is an essential issue to be considered, since the Fluoroprobe factory fingerprints are determined on mono-specific cultures or target micro-algae that are not necessarily representative to our shelf and ocean system (Houliez et al., 2012). No nutrient or

light measurements were taken either; therefore, we cannot explicitly describe any inter-species competition which will help us understanding the ecological consequences of these submesoscale processes. A distinct spectral community structure was anyway detected, compared to the surrounding waters, which could potentially be extended through the trophic web and even affect top predator's foraging behaviour (Cotté et al., 2015; Tew Kai et al., 2009). Thus, our results suggest, that the combined effects of submesoscale features, even though concerning a relatively small fraction of the total area, may be disproportionately

important to biological dynamics.

## 5   Conclusions

In the present study we conducted a joint analysis of operational data together with discrete data to describe submesoscale processes and the influence exerted on the distribution of the two major phytoplankton pigmentary groups in the SE-BoB. Satellite

imagery provides information about the surface-most layer, which is highly conditioned by the run-off of Adour and Bidasoa rivers. The location of the plume depends on the surface currents, which are ultimately conditioned by the direction of the wind.

Multi-spectral chl-a measurements allowed us to identify the contrasting effects of the environmental variables for the different phytoplankton spectral groups. From top to bottom, salinity explains most of the distribution of the chl-a for both

Brown and Green algae. While salinity would still be the most important environmental driver for Green algae at the DCM, vorticity explains most of the deviance of the distribution of Total chl-a and Brown algae chl-a at this layer. Anticyclonic circulation gathered the Brown algae in the center via Ekman transport. The effect is such that the distribution of Brown algae within the DCM cannot be statistically explained without the vorticity as an environmental variable. This research brings into consideration the relevance of the dynamic variables in the study of phytoplankton, as well as the measurements of

multi-spectral chl-a at high spatial resolution. Only by combining both we were able to determine the relative importance of the environmental variables for different spectral phytoplankton groups at the DCM. Further investigations providing a more detailed composition of the phytoplankton community in terms of pigments, size classes and taxonomy, together with an exhaustive analysis of the hydrodynamics, will help to better identify the ecological and functional traits of phytoplankton groups and determine their submesoscale distribution in coastal systems.

# Appendix A: Supplementary Material

## A1 Observation of eddies after ETOILE

After the change in wind regime on August $7^{th}$ the eddy C17W is again visible in the HF radar. Its has moved southwards with respect its location on July $29^{th}$. Meanwhile, C17E has vanished although it might be just masked by the surface currents since a meandering is still visible in its former location at $1.7^{o}$W.

## A2 Phytoplakton Observations at T3

A cross section at the 43.70 $^{o}$N out of the core of the anticyclonic frontal area, revels that this pattern is not ubiquous. Here there is not a clear dichotomy among the groups nor a deeper maximum of green algae. Rather, there is a uniform layer of brown algae.

*Author contributions.* XD, AR, FA, IP, IMN, PL and AC contributed to the main structure and contents. In addition, XD produced the figures, AR, FA, IMN, IP and PL participated in the measurements and IP and PL coordinated the cruise.

*Competing interests.* The authors declare that they have no conflict of interest.

*Acknowledgements.* This project was supported by the JERICO-NEXT (Joint European Research Infrastructure for Coastal Observatory – Novel European eXpertise for coastal observaTories) project within the European Union's Horizon 2020 research and innovation programme (grant agreement no. 654410). SST and Chl-a data are produced and distributed by NERC Earth Observation Data Acquisition and Analysis Service (NEODAAS, https://www.neodaas.ac.uk/Access_Data). We thank the Emergencies and Meteorology Directorate – Security department – Basque Government for public data provision from the Basque Operational Oceanography System EuskOOS. Ivan Manso-Narvarte was supported by a PhD fellowship from the Department of Environment, Regional Planning, Agriculture and Fisheries of the Basque Government. We would like to thank everyone who participated in the ETOILE Campaign and has collected or processed the data as well as the "cote de la Manche" crew.
This is the contribution number XXXX of the Marine Research Division of AZTI-Tecnalia.

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

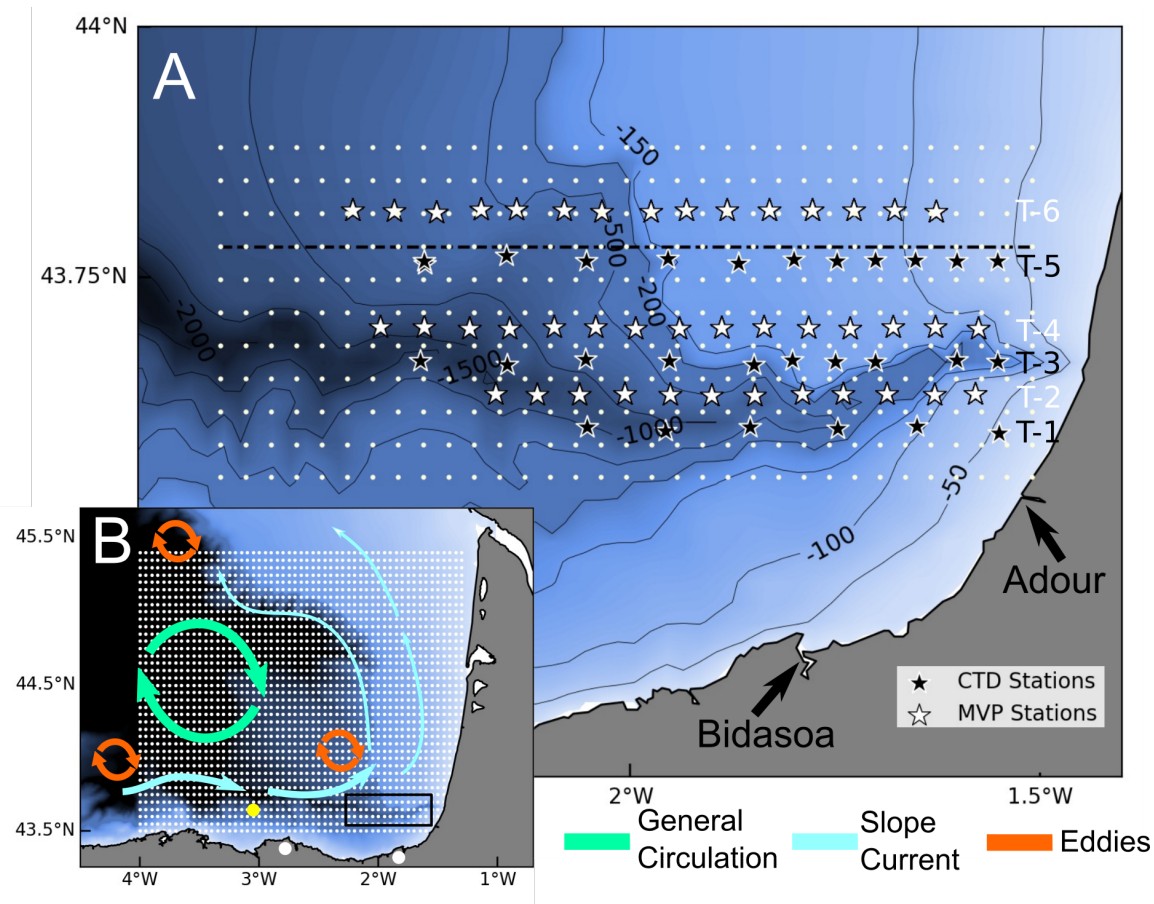

**Figure 1.** Sampling map and circulation in the Bay of Biscay. Locations of the CTD and Moving Vessel Profiler (MVP) stations are shown in A. At uneven transects (T-1, T-3 and T-5) black stars mark the CTD stations where vertical casts of temperature, salinity and in vivo multi-spectral chlorophyll-a (chl-a) were collected. At even transects, white stars mark the location of the point at which MVP data has been averaged, these are located every 5 km. White dots represent the grid at which these measurements were interpolated and the black dashed line mark the cross-section at 43.77ºN analysed in the Results Section. The location of the rivers Adour and Bidasoa is shown by the black arrows. The zoomed out map in the lower left corner (B) shows the circulation in the Bay of Biscay. The general circulation is characterized by a weak anticyclonic circulation in the central regions. Coastal areas are subjected to a seasonal slope current that, when interacting with the bathymetry results in the generation of eddies. The small white dots represents to the HF radar grid, the yellow dot corresponds to the location of the oceano-metereological buoy used for the wind data, while big white dots to the locations of the radar antennas. The black square shows the zoom in area shown in A.

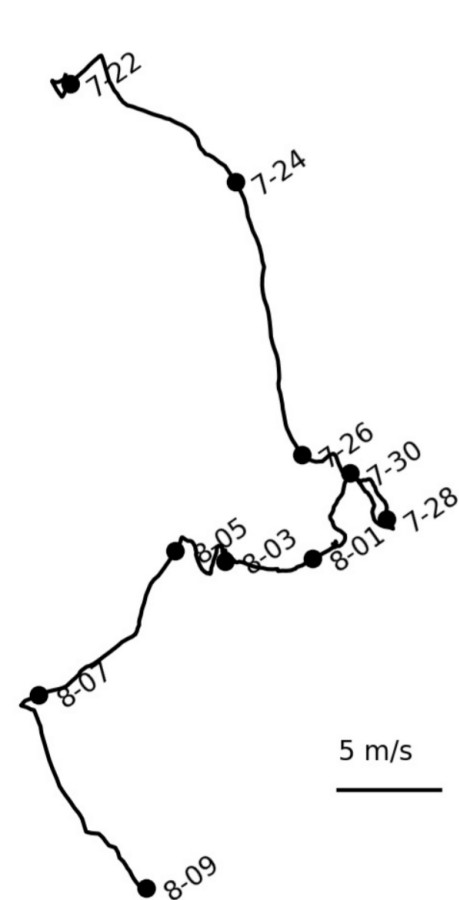

**Figure 2.** Wind direction and intensity at Bilbao's mooring buoy represented on a Progressive Vector Diagram (PVD)

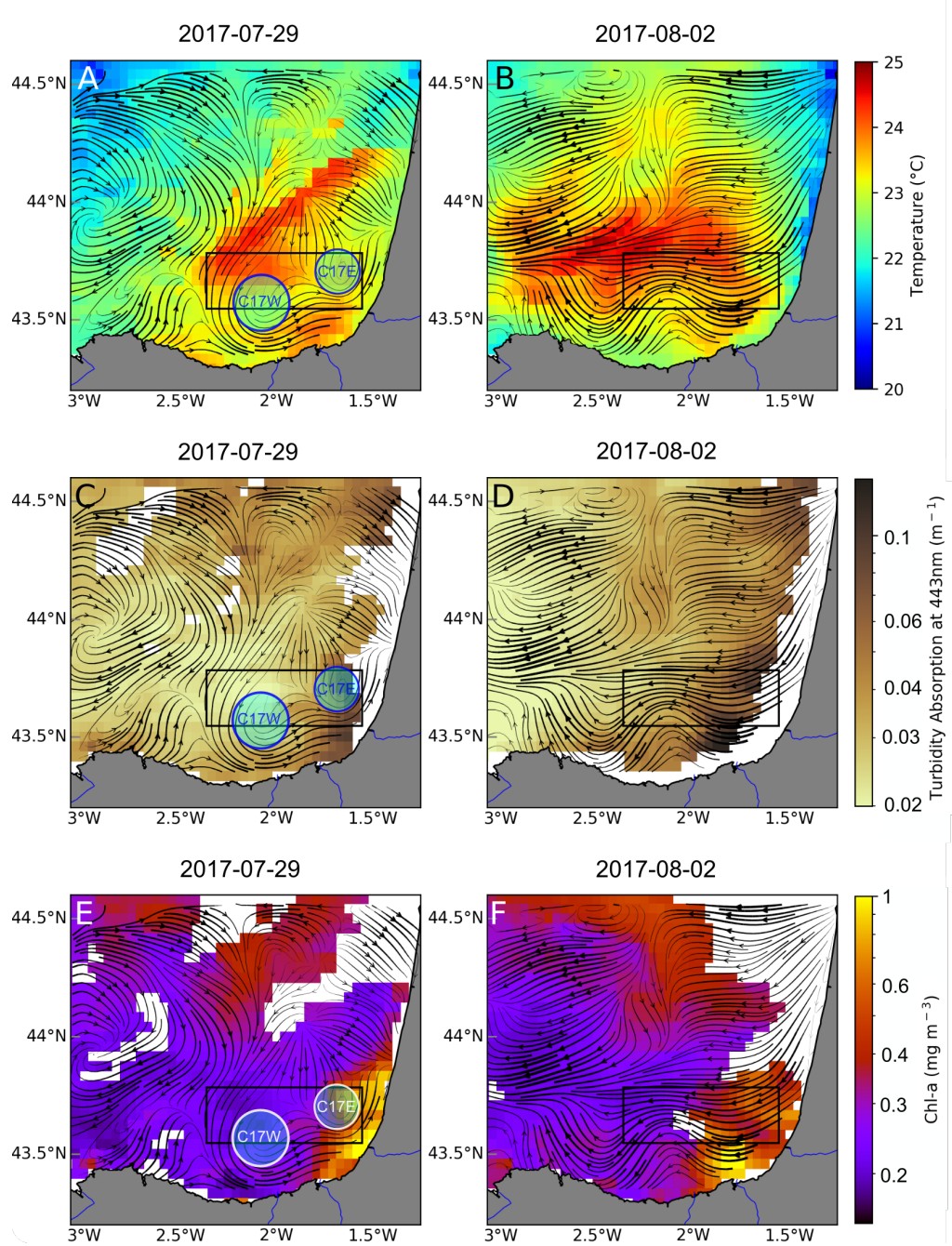

**Figure 3.** Hydrographic and hydrodynamical context. SST (A,B), turbidity (C,D) and chl-a (E,F) corresponding to July $29^{th}$ (left column) and August $2^{nd}$ (right column). Black lines show the LRC calculated for the previous three days, encompassing to periods; July 26-$29^{th}$ and July $30^{th}$ to August $2^{nd}$. The black box show the study area where the sampling took place. Turbidity and chl-a are plot on logarithmic scale

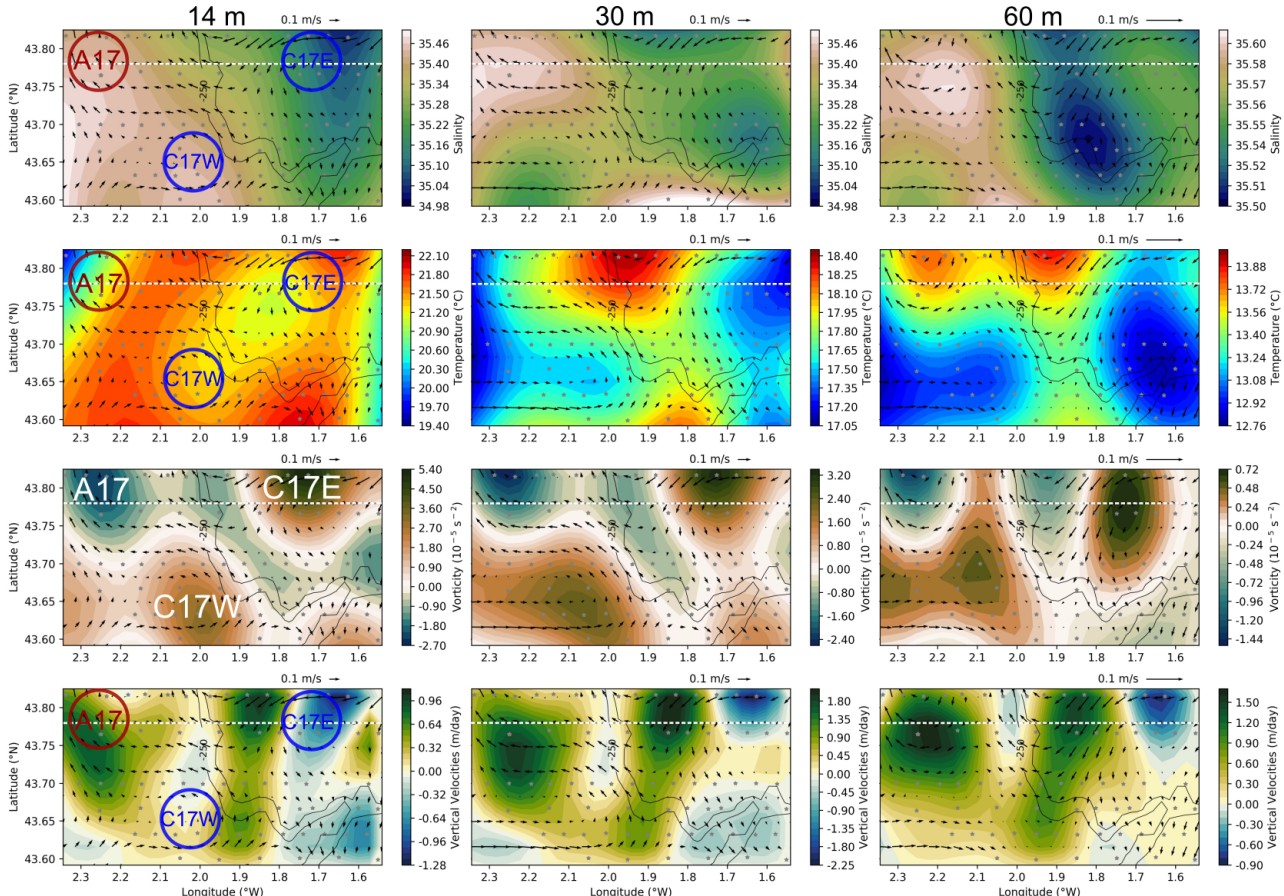

**Figure 4.** Synoptic composites for the hydrographic and hydrodynamical context obtained from the measurements during Etoile and representative for the same period (August $2^{nd}$ to $4^{th}$ 2017). From top to bottom: salinity, temperature, vorticity and vertical velocity fields. Each of the variables is mapped at 14, 30 and 60 m (left to right). Black arrows correspond to the geostrophic velocities and black contours represent the shelf break. The white dashed line corresponds to the cross section at $43.77^{o}$N. Negative (positive) vorticity values represents anticyclonic (cyclonic) circulation. Negative (positive) vertical velocity values represents downwelling (upwelling).

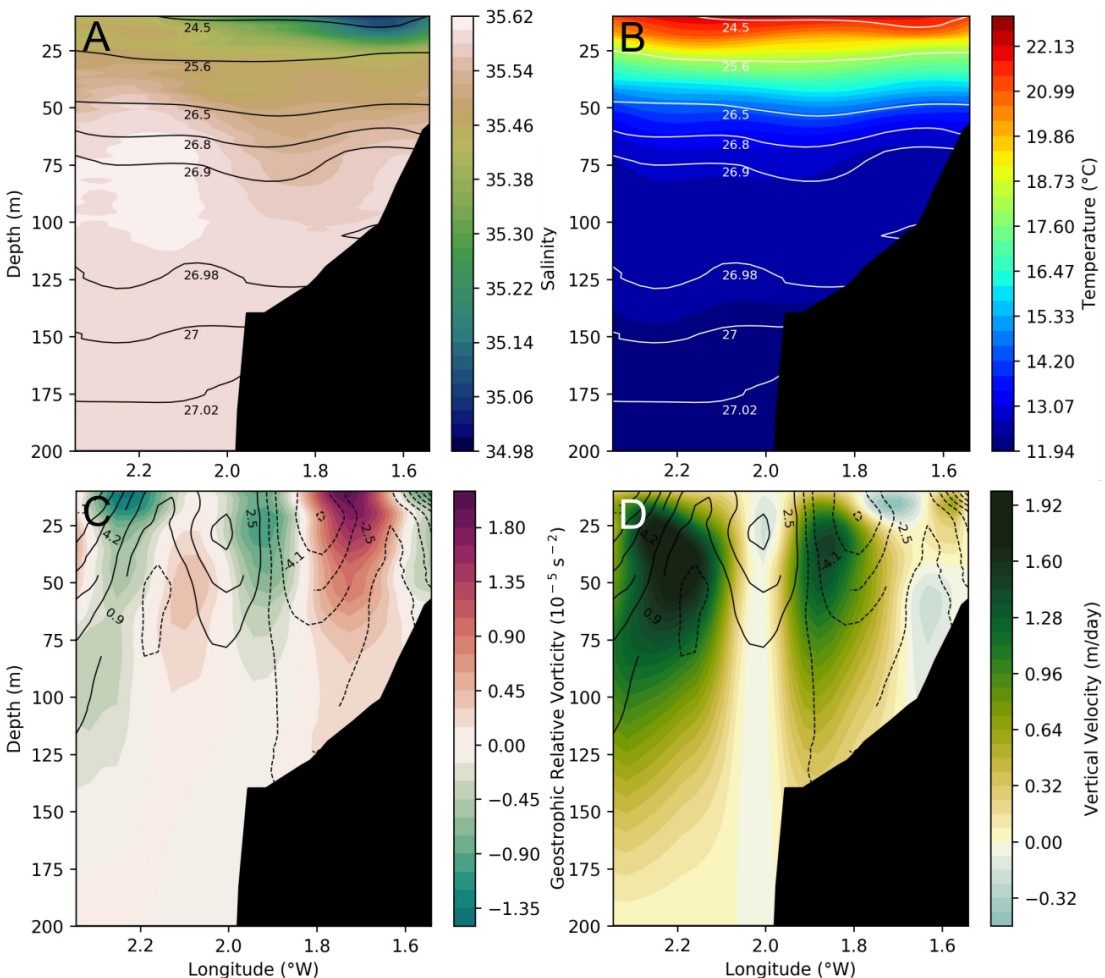

**Figure 5.** Cross section at 43.77°N (location marked by dashed lines in Figure 1 and 4), representing salinity (A), temperature (B) with isopycnals (black contours), vorticity (C) and vertical velocities (D) with meridional geostrophic velocities (black contours). Negative (positive) vorticity values represents anticyclonic (cyclonic) circulation. Positive (negative) values for geostrophic velocity represent northward (southward) current. Negative (positive) vertical velocity values represents downwelling (upwelling).

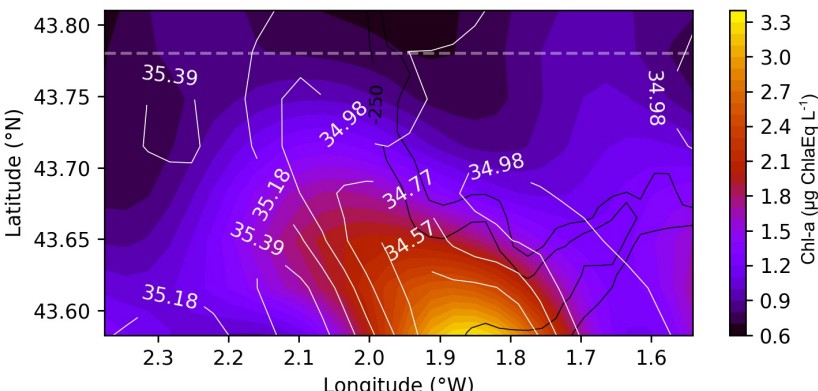

**Figure 6.** Surface chl-a (in chl-a Equivalent units) recorded by continuous measurements at 3.5 m, white contours represent the salinity field while the black the continental shelf.

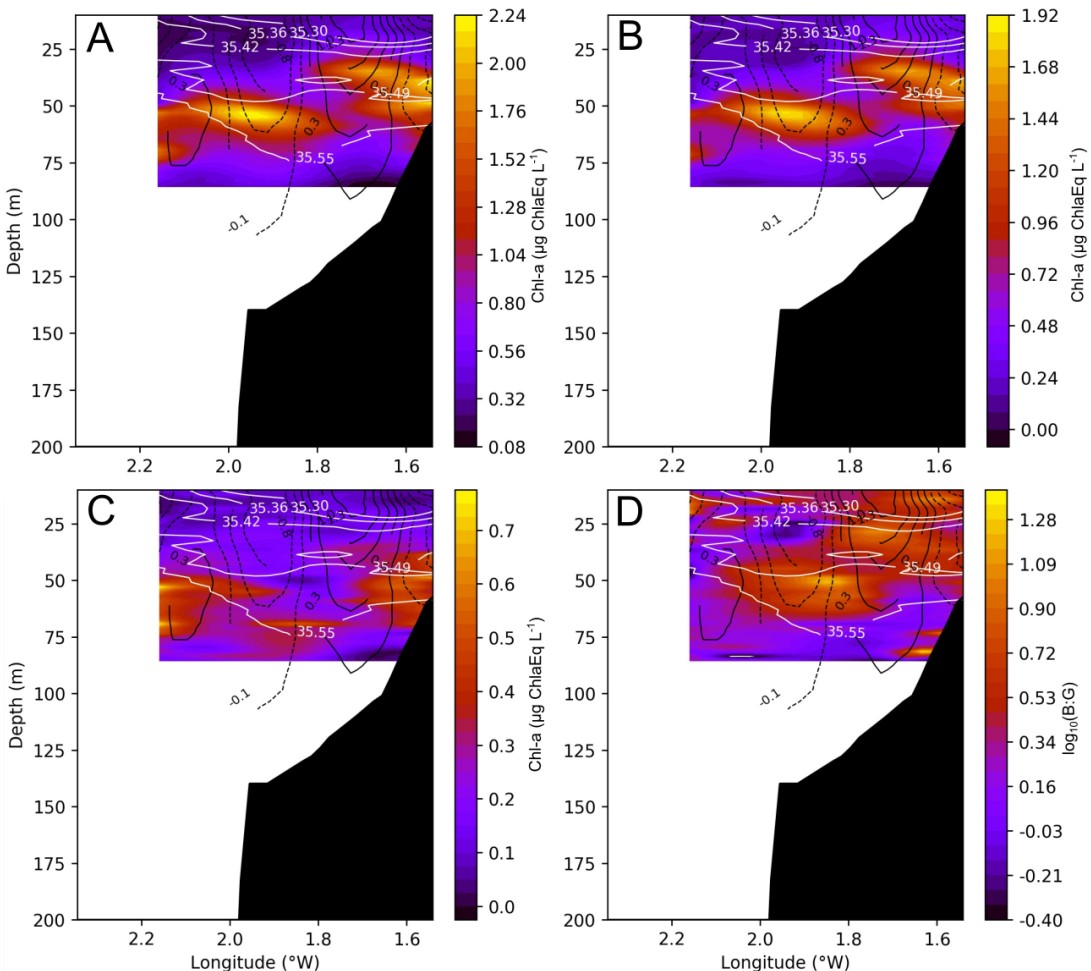

**Figure 7.** Cross section at 43.77$^o$N of Total chl-a(A), Brown algae chl-a (B), Green algae chl-a (C) and the Brown:Green ratio logarithmically normalized (D). White lines represent salinity contours and black solid (dashed) lines represent positive (negative) vorticity values or cyclonic (anticyclonic) circulation.

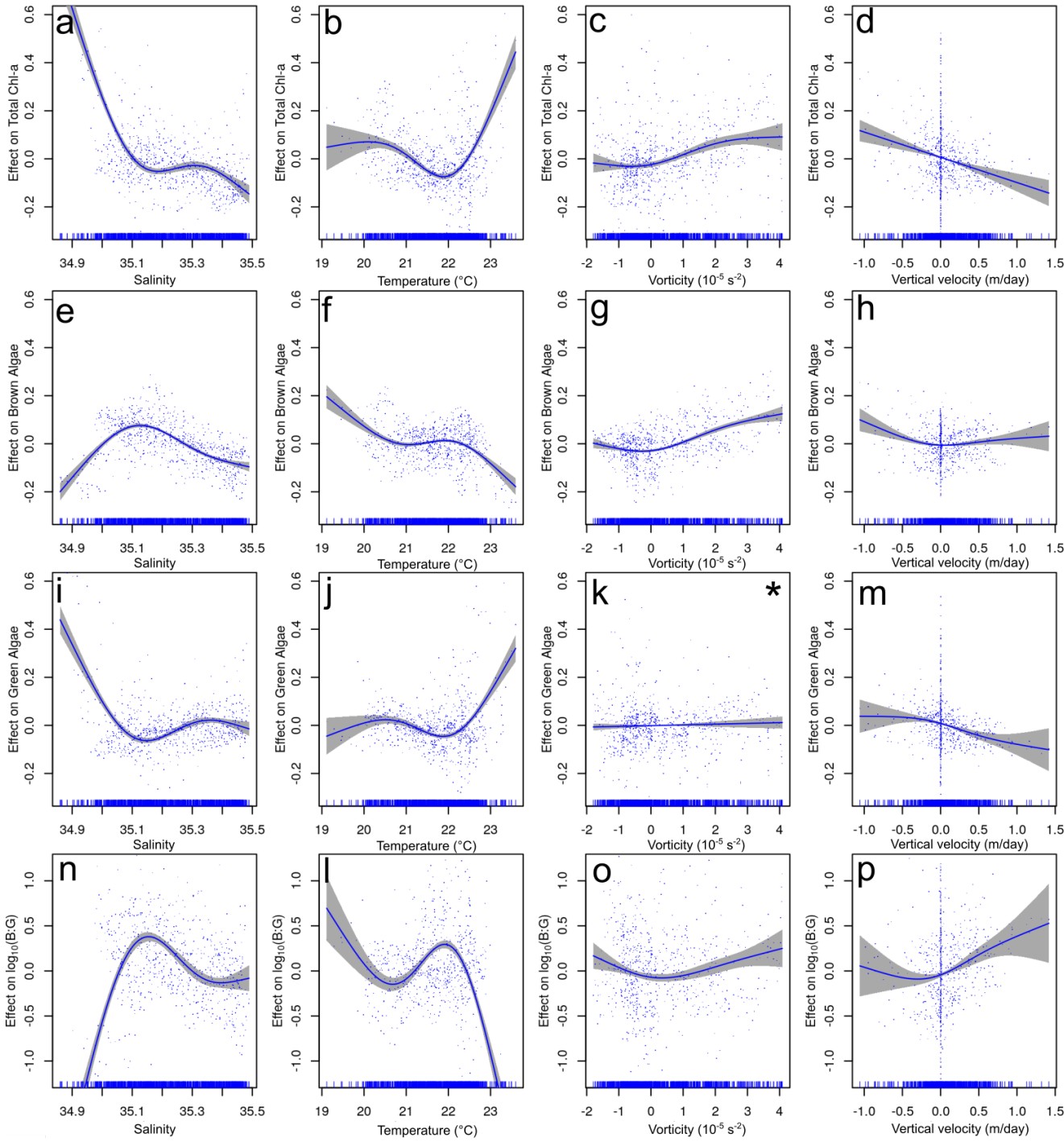

**Figure 8.** Relationships between environmental variables and chl-a from the models for the section above the pycnocline. The y-axis indicates the additive effect that the term on the x-axis has on the chl-a. From top to bottom, Total chl-a, Brown algae chl-a, Green algae chl-a and the Brown:Green ratio. Shaded area represents the confidence interval of 95%. The effect of vorticity for Green-algae chl-a is the only non-significant response (marked by *).

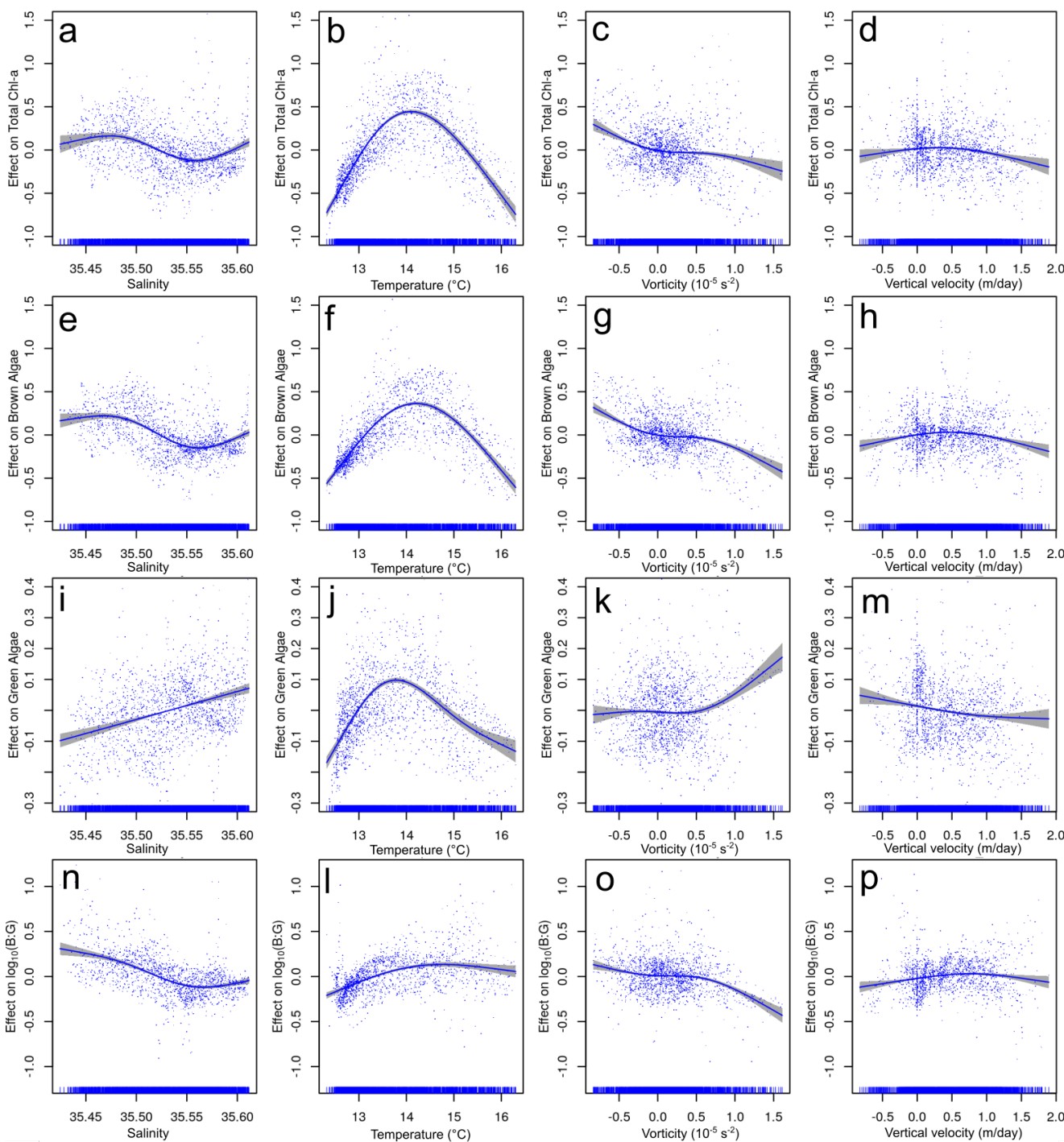

**Figure 9.** Relationships between environmental variables and chl-a from the models for the section below the pycnocline. The y-axis indicates the additive effect that the term on the x-axis has on the chl-a. From top to bottom, Total chl-a, Brown algae chl-a, Green algae chl-a and the Brown:Green ratio. Shaded area represents the confidence interval of 95%.

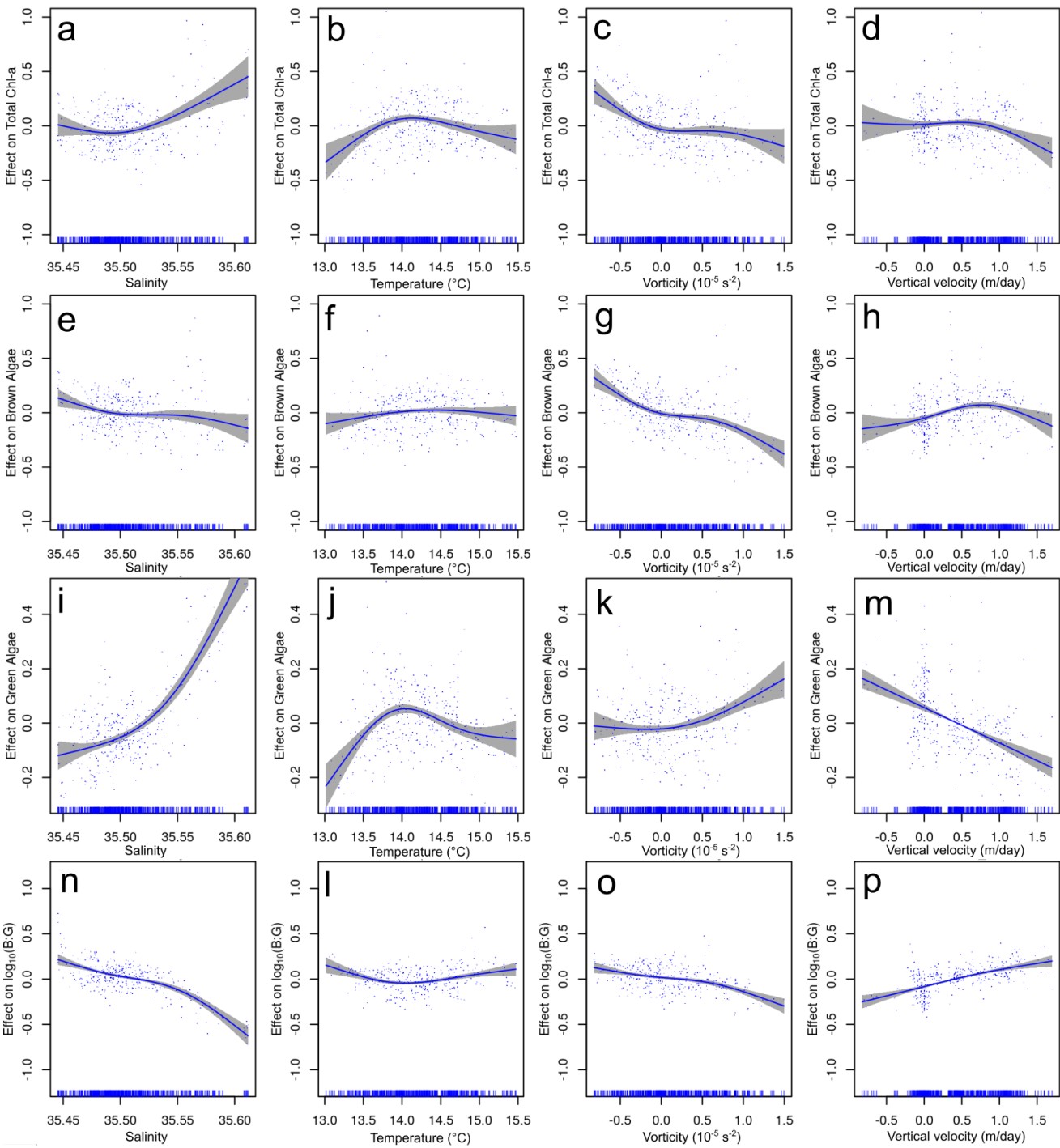

**Figure 10.** Relationships between environmental variables and chl-a from the models for the DCM. The y-axis indicates the additive effect that the term on the x-axis has on the chl-a. From top to bottom, Total chl-a, Brown algae chl-a, Green algae chl-a and the Brown:Green ratio. Shaded area represents the confidence interval of 95%.

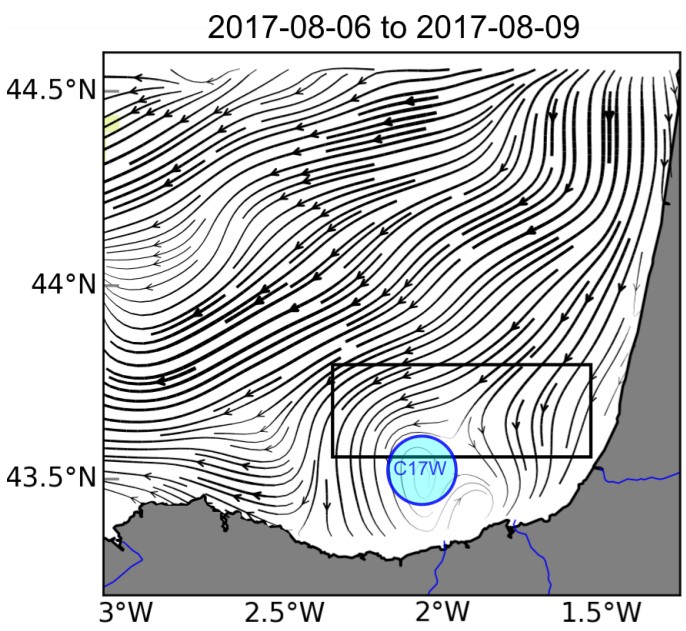

**Figure A1.** LRC for the period of August $6^{th}$ to $9^{th}$, the persistent C17W eddy is still visible after the change in wind regime.

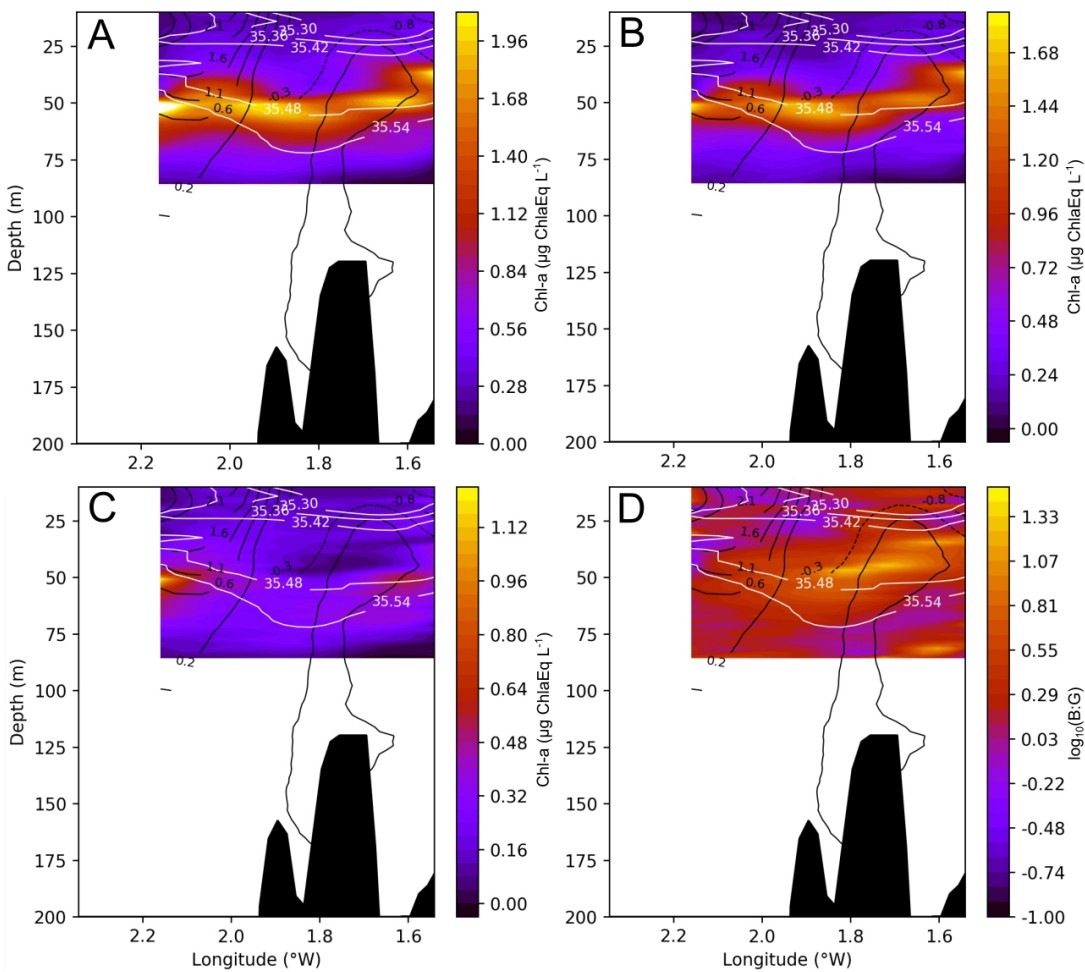

**Figure A2.** Cross section at 43.70$^o$N of Total chl-a(A), Brown algae chl-a (B), Green algae chl-a (C) and the Brown:Green ratio logarithmically normalized (D). White lines represent salinity contours and black solid (dashed) lines represent positive (negative) vorticity values or cyclonic (anticyclonic) circulation.

**Table 1.** Generalized Additive Model (GAM) results. Intercept, standard deviation (SE), significance (p-value) and explained devianced (%) of the GAMs for Above the pycnocline, Below the pycnocline and the Deep Chlorophyll Maximum (DCM). The estimated degrees of freedom (edf) and significance (p-value) of the environmental variables are also included. Although salinity and temperature were correlated for the section below the pycnocline, both variables were kept since the fit ($R^2$ and GCV) was better in all cases.

| | | Above | | Below | | DCM | |
|---|---|---|---|---|---|---|---|
| | | **Estimate** | **p-value** | **Estimate** | **p-value** | **Estimate** | **p-value** |
| **Total Chl-a** | Intercept | 0,380 | <0.001 | 1,096 | <0.001 | 1,793 | |
| | SE | 0,005 | | 0,00062 | | 12 | |
| | % | 60.8 | | 66 | | 17,3 | |
| | | **edf** | **p-value** | **edf** | **p-value** | **edf** | **p-value** |
| | Vertical_vel | 1 | <0.001 | 2,461 | <0.001 | 2,618 | 0,009 |
| | Temperature | 2,979 | <0.001 | 2,972 | <0.001 | 2,851 | <0.001 |
| | Vorticity | 2,788 | <0.001 | 2,896 | <0.001 | 2,744 | <0.001 |
| | Salinity | 2,990 | <0.001 | 2,974 | <0.001 | 2,484 | <0.001 |
| | | $R^2 = 0,603$ | GCV = 0,020 | $R^2 = 0,657$ | GCV = 0,069 | $R^2 = 0.148$ | GCV = 0,051 |
| **Brown Chl-a** | | **Estimate** | **p-value** | **Estimate** | **p-value** | **Estimate** | **p-value** |
| | Intercept | 0,206 | <0.001 | 0,775 | <0.001 | 1,374 | <0.001 |
| | SE | 0,003 | | 0,006 | | 0,00912 | |
| | % | 57.10 | | 71,8 | | 37,7 | |
| | | **edf** | **p-value** | **edf** | **p-value** | **edf** | **p-value** |
| | Vertical_vel | 2,658 | <0.001 | 2,65 | <0.001 | 2,844 | <0.001 |
| | Temperature | 2,988 | <0.001 | 2,981 | <0.001 | 2,025 | 0,09 |
| | Vorticity | 2,934 | <0.001 | 2,96 | <0.001 | 2,816 | <0.001 |
| | Salinity | 2,983 | <0.001 | 2,97 | <0.001 | 2,596 | 0,004 |
| | | $R^2 = 0,564$ | GCV = 0,051 | $R^2 = 0,717$ | GCV = 0,044 | $R^2 = 0.359$ | GCV = 0,030 |
| **Green Chl-a** | | **Estimate** | **p-value** | **Estimate** | **p-value** | **Estimate** | **p-value** |
| | Intercept | 0,148 | <0.001 | 0,321 | <0.001 | 0,418 | <0.001 |
| | SE | 0,004 | | 0,003 | | 0,0058 | |
| | % | 43 | | 34,1 | | 56,9 | |
| | | **edf** | **p-value** | **edf** | **p-value** | **edf** | **p-value** |
| | Vertical_vel | 2.353 | <0.001 | 1,924 | <0.001 | 1 | <0.001 |
| | Temperature | 2,983 | <0.001 | 2,988 | <0.001 | 2,978 | <0.001 |
| | Vorticity | 1.000 | 0,362 | 2,871 | <0.001 | 2,263 | <0.001 |
| | Salinity | 2.986 | <0.001 | 1 | <0.001 | 2,784 | <0.001 |
| | | $R^2 = 0,423$ | GCV = 0,013 | $R^2 = 0,337$ | GCV = 0,012 | $R^2 = 0.558$ | GCV = 0,012 |
| **B:G** | | **Estimate** | **p-value** | **Estimate** | **p-value** | **Estimate** | **p-value** |
| | Intercept | 0,109 | <0.001 | 0,36 | <0.001 | 0,543 | <0.001 |
| | SE | 0,019 | | 0,004 | | 0,006 | |
| | % | 55 | | 57,2 | | 64,5 | |
| | | **edf** | **p-value** | **edf** | **p-value** | **edf** | **p-value** |
| | Vertical_vel | 2,452 | <0.001 | 2,588 | <0.001 | 1,866 | <0.001 |
| | Temperature | 2,996 | <0.001 | 2,712 | <0.001 | 2,874 | <0.001 |
| | Vorticity | 2,572 | 0,056 | 2,941 | <0.001 | 2,652 | <0.001 |
| | Salinity | 2,983 | <0.001 | 2,935 | <0.001 | 2,819 | <0.001 |
| | | $R^2 = 0,544$ | GCV = 0,283 | $R^2 = 0,569$ | GCV = 0,034 | $R^2 = 0,635$ | GCV = 0,015 |

**Table 2.** Deviance contribution of the environmental variables. Coefficient of determination ($R^2$), general cross validation score (GCV) and percetage of variance (%) accounted by the different models after deletion of one variable. The left columns in each of the sections show these values for the models after stepwise deletion of the variables listed to the left (first vertical velocities and then temperature). The last two models included only the variable listed (vorticity and salinity). For the right columns in each section, one variable (those listed on the left) was removed at a time while keeping the rest.

| | | Above | | | | | | Below | | | | | | DCM | | | | | |
| | | Stepwise Deletion | | | Delete-one-covariance | | | Stepwise Deletion | | | Delete-one-covariance | | | Stepwise Deletion | | | Delete-one-covariance | | |
| Group | Variable | $R^2$ | GCV | % | $R^2$ | GCV | % | $R^2$ | GCV | % | $R^2$ | GCV | % | $R^2$ | GCV | % | $R^2$ | GCV | % |
|---|---|---|---|---|---|---|---|---|---|---|---|---|---|---|---|---|---|---|---|
| Total Chl-a | All (total) | 0,600 | 0,020 | | | | | 0,66 | 0,069 | | 0,652 | 0,07 | | 0,15 | 0,052 | | | | |
| | Vertical_vel | 0,590 | 0,021 | 1,40 | 0,590 | 0,021 | 1,40 | 0,652 | 0,070 | 34,20 | 0,32 | 0,14 | **34,00** | 0,12 | 0,055 | 3,40 | 0,12 | 0,05 | 3,40 |
| | Temperature | 0,460 | 0,027 | 58,91 | 0,510 | 0,025 | 9,80 | 0,32 | 0,138 | 59,62 | 0,64 | 0,07 | 2,10 | 0,08 | 0,057 | 7,88 | 0,09 | 0,06 | 6,30 |
| | Vorticity | 0,020 | 0,077 | 11,50 | 0,580 | 0,022 | 2,50 | 0,06 | 0,184 | 35,50 | 0,64 | 0,07 | 2,20 | 0,05 | 0,058 | 11,07 | 0,06 | 0,06 | **9,97** |
| | Salinity | 0,460 | 0,027 | 14,40 | 0,470 | 0,042 | **13,10** | 0,30 | 0,140 | | | | | 0,02 | 0,060 | 15,45 | 0,07 | 0,06 | 8,98 |
| Brown Chl-a | All (total) | 0,564 | 0,005 | | | | | 0,72 | 0,044 | | | | | 0,36 | 0,031 | | | | |
| | Vertical_vel | 0,553 | 0,005 | 1,20 | 0,553 | 0,005 | 1,20 | 0,71 | 0,046 | 0,70 | 0,71 | 0,05 | 0,70 | 0,32 | 0,033 | 4,40 | 0,32 | 0,03 | 4,40 |
| | Temperature | 0,464 | 0,006 | 10,30 | 0,490 | 0,006 | 7,50 | 0,44 | 0,088 | 27,70 | 0,44 | 0,09 | **27,30** | 0,30 | 0,033 | 6,60 | 0,35 | 0,03 | 0,70 |
| | Vorticity | 0,288 | 0,008 | 28,00 | 0,441 | 0,007 | 12,40 | 0,09 | 0,143 | 63,11 | 0,68 | 0,05 | 4,00 | 0,26 | 0,035 | 11,10 | 0,17 | 0,04 | **19,30** |
| | Salinity | 0,135 | 0,010 | 43,30 | 0,331 | 0,008 | **23,30** | 0,42 | 0,091 | 29,70 | 0,63 | 0,06 | 8,80 | 0,08 | 0,043 | 28,69 | 0,34 | 0,03 | 1,60 |
| Green Chl-a | All (total) | 0,423 | 0,013 | | | | 0,337 | 0,01 | 0,034 | | | | | 0,56 | 0,012 | | | | |
| | Vertical_vel | 0,404 | 0,013 | 2,00 | 0,404 | 0,013 | 2,00 | 0,33 | 0,012 | 1,3 | 0,33 | 0,01 | 1,3 | 0,46 | 0,015 | 10,5 | 0,46 | 0,02 | 10,5 |
| | Temperature | 0,300 | 0,015 | 12,70 | 0,320 | 0,015 | **10,40** | 0,13 | 0,015 | 20,5 | 0,14 | 0,02 | **19,90** | 0,43 | 0,016 | 13 | 0,50 | 0,01 | 6,00 |
| | Vorticity | 0,011 | 0,044 | 41,64 | 0,423 | 0,0125 | 0,00 | 0,04 | 0,017 | 30,15 | 0,64 | 0,07 | 2,10 | 0,21 | 0,022 | 36,20 | 0,52 | 0,01 | 4,10 |
| | Salinity | 0,300 | 0,015 | 12,70 | 0,427 | 0,026 | 0,00 | 0,01 | 0,016 | 24,56 | 0,31 | 0,01 | 2,80 | 0,37 | 0,017 | 19,20 | 0,30 | 0,02 | **25,40** |
| B:G | All (total) | 0,544 | 0,283 | | | | | 0,57 | 0,034 | | | | | 0,64 | 0,015 | | | | |
| | Vertical_vel | 0,530 | 0,289 | 1,30 | 0,530 | 0,289 | 1,30 | 0,56 | 0,035 | 1,00 | 0,56 | 0,03 | 1,00 | 0,50 | 0,020 | 14,00 | 0,50 | 0,02 | 14,00 |
| | Temperature | 0,373 | 0,385 | 17,30 | 0,385 | 0,379 | **15,80** | 0,51 | 0,038 | 5,80 | 0,53 | 0,04 | 4,50 | 0,50 | 0,020 | 14,20 | 0,60 | 0,02 | 3,40 |
| | Vorticity | 0,020 | 0,657 | 52,46 | 0,535 | 0,287 | 1,00 | 0,08 | 0,721 | 48,58 | 0,53 | 0,04 | 4,20 | 0,30 | 0,028 | 34,10 | 0,54 | 0,02 | 9,30 |
| | Salinity | 0,631 | 0,392 | 18,70 | 0,387 | 0,414 | 15,60 | 0,48 | 0,409 | 9,10 | 0,45 | 0,04 | **11,80** | 0,32 | 0,027 | 31,70 | 0,43 | 0,02 | **20,00** |