# Peer review of "Coastal submesoscale processes and their effect on phytoplankton distribution in the SE Bay of Biscay"

_Ocean Science, 2020_

## Author Comment (AC1) · 18 Jun 2020

An old version of the figure was uploaded. The relation between the vorticity and the brown:green algae ratio for the section "above the pycnocline" is now marked with an asterisk, as not significant.

---

## Referee Comment (RC1) · Anonymous Referee #1 · 31 Jul 2020

The manuscript describes the occurrence of different (sub)mesoscale processes during a summer in the SE Bay of Biscay and their effect on phytoplankton distribution. This work is based on a comprehensive data set collected at different spatiotemporal scales and the research topic is interesting. However, there are many issues that have to be addressed in order to be published.

General comments:

1. The structure, organization and narrative of the paper are difficult to follow, especially during the introduction and discussion (see comments below). I think this has to be deeply revised. In addition, sometimes the paper seems to be written in haste,

with several grammar, spelling and punctuation problems (see a few examples in the technical comments). Also, check things such as "(?)" in L402 and 427.

2. Throughout the entire paper, there are several assertions without references (see comments below). For instance, L18-19 or L22-23 in the introduction.

3. In the introduction, the motivation of the study, the relevance of the topic and knowledge gap(s) have to be better introduced and established. For instance, I find interesting studying the effect of (sub)mesoscale processes in coastal areas due to its complexity, and maybe there are few studies about it in the BoB. However, L58: "Few studies provide the link between phytoplankton occurrence and physical processes in the BoB" is definitely not true. In relation to this, maybe it could be highlighted and argued the importance of using data collected at different spatiotemporal scale (something this paper does well). Also, I do not feel that the introduction goes from general to more particular aspects of the research topic. In some paragraphs, there are too many ideas (although there should 1 or 2), which is especially clear in paragraph 1 (L18-29). Why not using a whole paragraph (or even 2 paragraphs) to develop the importance of (sub)mesoscale processes and phytoplankton and their link? Why focusing so fast on eddies when this is just one case? Why introducing already the BoB? Other paragraphs just feel like a collection of examples with no clear message (for instance L58-67: paragraph 4). Additionally, the ideas and concepts have to be connected better, within and among paragraphs.

4. The statistical analyses should be better conducted. This also affects section 3.3 in results. Although I like the use of GAMs, I really miss a model comparison based for example on AIC (or other criteria). Why are predictors inspected only one by one? What about interaction effects among predictors? In GAMs, those could be included as tensor products or varying coefficient models. Have the authors checked if the predictor variables are correlated? I think using a model with interaction is a better approach than the analyses described in L160-162 and L339-348. Also, in material and methods, the description and specifications of the model(s) and statistical methodology

is too scarce. What are the formulas and specifications for the model(s) used? What are the characteristics of the residuals? In results, what about deviance or variance explained by the models?

5. Sometimes, the study mixes different biological concepts. For instance, fluorescence can be translated to Chl a concentration. However, Chl a concentration is not the same as phytoplankton biomass, as this depends on variations of C:Chl a cell ratios. In addition, larger biomass accumulation does not necessarily imply larger phytoplankton growth. See for instance L263 and L361.

6. In my opinion, the authors should translate better the relationships found into a mechanistic understanding. How could salinity affect distribution? Is it directly affecting phytoplankton physiology and growth? Is it because differences in salinity reflect the occurrence of processes such as fronts or other modifications of the physical structure of the water column? The same goes for vorticity.

7. In my opinion, the discussion is the weakest part of the article. It has to be better structured and the implications more clearly defined at the end (including the conclusions). Please, be more concrete, thinking about how specifically does this study contribute to the field (the statements at the end of the conclusion sounds too generic).

Specific comments:

Abstract

L2-4: Why just focusing on the effect on nutrients (which are not directly analyzed in the study)? What about the modification of the water column structure? They only define niches or also affect phytoplankton through advection? I like the motivation for the study in L4-5.

L11: This is again the goal of the study. Merge it with L6-7.

L15-16: Do studies analyze hydrographic aspects and not consider the dynamics of the system? Think about more particular and specific implications of this work.

Introduction

L18: Please define spatiotemporal scale ranges for mesoscale processes (km and days). For instance, Mahadevan (2016) in L22 actually talks about submesoscale processes. Maybe mesoscale should be replaced by submesoscale in the entire article (or at least in some parts)?

L20: Not only nutrients can be limiting. Light is the other main factor (e.g. in temperate and polar areas in winter or at depth). Elaborate.

L20: Are not all processes confined in time and space? What does this mean?

L21: After "evolve and transport seawater properties" there is a "-i.e. nutrients". However, this is just one of the non-conservative properties.

L31-40: Give a purpose to the paragraph by highlighting that the study area is complex in terms of coastal hydrographic processes.

L44: Include a mention to a figure in the introduction is unusual as far as I know.

L76-78 should be integrated with L71-72. Both parts are about what was done in the study.

Material and Methods

L90-102: Include references for all the instrument and methodology used. Also, is there any calibration? Chlorophyll is mentioned here, but fluorescence is used for the analyses instead. Four algal pigmentary groups can be detected by the fluorometer, but only 2 are inspected in this study. Why? If there were an instrumental bias and different measuring sensitivity for the 2nd multi-spectral fluorometer, can the data be trusted at all?

L113: Why a 3h running mean average was applied? Please explain and justify.

L135: Define small Rossby number range. Is it realistic to assume totally constant mesoscale features during the sampling? If not, elaborate this. Also, replace time/space by spatiotemporal. In relation to this point, is it not always important the spatiotemporal distribution of observations?

L139: Which are those key dynamical variables? Please elaborate.

L148: "the correct representation" is a strong statement. Better "good" or "appropriate".

L153: Include the statistical analyses as a different section with a title (2.4).

L159: Please indicate the version of R and mgcv package used. Also, cite the R program.

Results

L166-168: Part of it belongs to Material and Methods or even to the introduction.

L171: "relaxed" is in my opinion vague.

L186: Do these results belong to the 1st or 2nd period? Clarify this.

L219-220, L222-224, L246-256 and L266-271 and L288-293: Belongs to the discussion.

L238: It is hard for me to observe how green algae fluorescence follow the salinity contours at waters saltier than 35.49. Please clarify this.

L239: "logarithmically transformed" sounds better to me.

L265-266: "isohaline" instead of "halocline"?

L266-271 and L288-293: Check the writing and description of the relationships (for instance, I do not observe a positive relationship at the edges of the range of temperature).

Discussion

L260-307: I think the first paragraph should be a brief summary of the main results.

L301-305 belong part to Material and Methods and part to Results.

L318: What do non-linear terms mean here? Join this paragraph with the previous one?

L327-328: Is this statement from other studies? Then, please include references. If not, from the results provided in this work, this can only be speculated (nutrients were not analyzed here). Riverine plumes have also other effects such as advect phytoplankton or generate fronts were plankton can accumulate.

L336-337: Again, this should be more speculative.

L339: See general comment 6. Include references.

L340-348: This belongs to Material and methods (and part to Results).

L350-365: This belongs to Results.

L372: What does mean "areas of vertical velocities"? of maximum velocities? Why should we expect higher concentrations in these areas? Elaborate.

L382-383: I do not get this sentence.

L409: Include references. Also, diatoms have different mechanisms to regulate their vertical position. This can be discussed too.

L418-430: I miss some references here.

Figures

-Include in the caption all the information necessary to understand each figure. For instance, in Fig. 1 indicate that the dot corresponds to the buoy, stars correspond to radar antennas and names to rivers (do the same for the other figures). Also, before the acronyms such as MVP, include the complete name.

-Figs. 1 and 2 can be joined in a single figure.

-Fig. 1: Are the eddies shown a permanent part of the general hydrography? Clarify.

-Fig. 2: Replace "T-" by: (T-1, T-3 and T-5).

-Fig. 3: Include axis with units.

-Fig. 4: Change labels of facets by A, B; C, D; E, F. Please, do not use scientific notation for the colour scales in this case. Also, note if the scales are logarithmic for turbidity and Chl a.

-Fig. 5: indicate in the caption that scale ranges are different for each depth. Salinity has no units, so delete PSU (apply this to other figures and text too). To what date(s) correspond(s) the maps? Indicate what positive and negative vertical velocities mean. The 43.77°N dashed lines are hard to see.

-Fig. 6: Are isolines actually isopycnals?

-Fig. 8: To which dates correspond the plots?

-Figs. 9 and 10: Caption should start with "Relationship between XXX and YYY". Please, do not use the default R output and replace variable names in the x-axis by the name of the variable and the units. Why the y-axis (fluorescence) in the 1st 3 rows can be negative?

-Fig. A2: Why is the cross-section here 43.70°N and not 43.77? Is it because C17W moved?

Technical corrections:

L33: Insert "of" before "the water runoff".

L116: Delete "of the".

L149: After "Gomis et al. (2001)" something is missing (in? by? As in?).

L180: Replace "generate" with "generated".

L182: Replace "; as well as" by a comma.

L196: Erase the 1st "wind".

L201: Include "at" after "(A17)". Also, replace "however" by "although".

L238: Replace "is" by "are".

L368: Replace the comma after "at first" by "as" or similar.

L486: Which number is "XXX"?

---

## Referee Comment (RC2) · Anonymous Referee #2 · 10 Aug 2020

General comments

The manuscript describes mesoscale processes in the shelf of the Southern Bay of Biscay and tries to relate that physical environment with the occurrence and distribution of phytoplankton in the area. The approach presented is very interesting and the manuscript provides a detailed description of a snapshot of the circulation in the SE BoB in August.

I have to acknowledge that I am not an expert in ocean circulation, so although I found this part well described and thoughtful, I am not fully capable of reviewing the methodological details of the description of the mesoscale ocean processes.

Since my expertise includes the phytoplankton community of the BoB, my main concerns are related to the fact that the aim of the manuscript is to relate the physical environment to the phytoplankton community structure, and I found this connection poorly supported by the data presented.

First, phytoplankton distribution is presented though accessory pigments fluorescence data, which is variable depending on the proportion of accessory pigments with respect to chlorophyll and depending on the proportion of chlorophyll to phytoplankton carbon. I think these fluorescence data do not represent phytoplankton distribution as straightforward as the authors claim. Also, not all phytoplankton groups are presented in the results, only "green" and "brown" algae, which leaves out all the cyanobacteria, very relevant in the phytoplankton community of the BoB in summer.

Regarding writing and composition, the manuscript is a bit difficult to follow, the physical part is better explained (although there are some typos and acronyms not defined, listed below), but the biology part is very confusing, with many concepts not fully explained.

Specific comments

Introduction

25 That's an unclear sentence, it is not clear which is the subject (it?) of the first part.

27 "this cross-self transport" does refer to the complex ocean dynamics mentioned before (26)?

64 Some word is missing here: "different phytoplankton groups" or "different groups of phytoplankton".

74 MFSD not defined.

Methods

90 Is the FluoroProbe deployed together with the CTD casts?

95 Typo: "Cryoyptophytes" should read Cryptophytes.

138 I think there is a typo here: "enough resolution for resolving".

149 Another typo, a parenthesis or a preposition is missing: "of the analysed field (Gamis et al. 2001)".

150 The treatment of fluorescence data is not very well explained. Only the method to interpolate the values to a regular grid is explained. But regarding the FluoroProbe data themselves, if FluoroProbe provides Chla values (95) why are they not showed and it is instead fluorescence? Are the fluorescence values calibrated with filtered samples in any way? Even though chlorophyll is not the same as phytoplankton biomass (given the variability in the chlorophyll to carbon ratios), it is more interpretable and comparable among groups than fluorescence. Fluorescence is also variable depending on the content of accessory pigments which is also subjected to photoacclimation and hence variable with phytoplankton physiological state. That's for me the weakest point of the manuscript, that the fluorescence values presented hardly represent the actual biomass or abundance of the phytoplankton community.

160 This methodology is not very clear, "smaller subsets in relation to the fluorescence" do refer to the spectral groups retrieved by the FluoroProbe?

Results

182 Correct punctuation: "the distribution of the SST, as well as the position of the river plumes".

Figure 5 (186) It seems the names of the eddies are duplicated in panels.

Figure 7 (226) Please, indicate which are the units for fluorescence, even if they are arbitrary units.

Figure 8 (230) Why values of total and groups of phytoplankton are not given in chlorophyll if the output of FluoroProbe is equivalent chlorophyll (95)? Maybe explaining the

FluoroProbe technique with more detail would help with the interpretation of the data, or at least including some references about the technique.

246 Figure 8 shows depth profile not surface fluorescence, maybe the text should read: "From satellite imagery and continuously recorded surface salinity and fluorescence data (Figure 4 and 7)".

Discussion

325 I don't think this sentence is correct. Which varies depending on the position in the water column could be which physical driver affects most the occurrence or distribution of phytoplankton, but not the interplay between physics and phytoplankton in general.

333 The authors seem to insist throughout the manuscript on the role of salinity/freshwater as one of the main drivers of the distribution of phytoplankton above the picnocline, which is more likely an effect of nutrient-availability (river discharge related). I would suggest the authors to take care of these kind of sentences that relate so directly salinity and phyto distribution.

339-348 This paragraph seems methods to me, not results. Maybe could be useful to have this paragraph in the methods section where the filtering technique is introduced to help explain its relevance (160), which is not very clear (see below).

350 I don't quite understand the point of this filtering technique. If I understood correctly with each iteration only the larger values are selected, and regarding chlorophyll this eventually considers only the large values in the DCM. But, with larger values correlation coefficients are also larger, not necessary meaning a higher correlation among data, so I am not sure that correlation coefficients between iterations are comparable. Also, with each iteration sample size, range and probably also variability are smaller which influences the comparability of correlation coefficients among iterations. I would suggest the authors to clarify the relevance of this statistical analysis.

353 "The strong negative correlation points suggest that in general brown algae are

highly conditioned by the salinity range". Conditioned by the salinity range in which sense?

372 Data presented are not of phytoplankton concentration.

403 The variable fluorescence to chlorophyll ratios could amplify or decrease the signal depending on if the fluorescence comes from accessory photosynthetic pigments (that increase relative to chlorophyll with depth) or from accessory photoprotective pigments (that decrease relative to chlorophyll with depth).

409 "The latter (dinoflagellates) can easily regulate their optimum depth by altering their swimming behaviour." Not sure about that, dinoflagellates can swim but not at the spatial scale necessary to change their position in the water column, working against turbulence, mixing and so on. If I am wrong, the authors should include some reference for this statement.

427 Possible references for fluorescence to chlorophyll ratios and for fluorescence fingerprint variable within groups and within populations: Estrada, Marrasé and Salat. In vivo fluorescence/chlorophyll a ratio as an ecological indicator in oceanography. Sci. Mar. (1996) 60(1) : 317-325. Kruskopf and Flynn. Chlorophyll content and fluorescence responses cannot be used to gauge reliably phytoplankton biomass, nutrient status or growth rate. New Phytologist (2006) 169: 525–536.

429 "In any case, vorticity creates a dynamical niche that plays a major role shaping the phytoplankton community". I find this is a too ambitious sentence, the "shape" of the phytoplankton community is not fully addressed in the manuscript and hence the major role of these vorticity-created niches has not been really evaluated.

435 This last paragraph is a mix of many concepts, phytoplankton functional types, biogeochemical models, harmful algae, fisheries... I would suggest to reorganize it and focus more clearly on the aims and findings of the manuscript.

Conclusions

447 "... joint analysis of remote and operational together with discrete data..." is confusing. Maybe repeat data after remote and operational.

---

## Author Comment (AC2) · 3 Nov 2020

**Dear reviewer,**

**Thank you for your thorough and critic review of our manuscript. Reading your comments, we have realized that the organization and structure of the manuscript was not easy to follow, neither the statistical analysis was detailed enough. Therefore, we have made significant changes throughout all the manuscript to correct these aspects and improve the readability of the paper. We hope that thanks to your suggestions we have managed to improve the manuscript, and that it suits now the standards of Ocean Science. The specific responses to your comments and the related changes are detailed in the following.**

**Best regards,**

**Xabier Davila**

AR = Author's response
AC = Author's changes in the manuscript

**General responses:**

*1. The structure, organization and narrative of the paper are difficult to follow, especially during the introduction and discussion (see comments below). I think this has to be deeply revised. In addition, sometimes the paper seems to be written in haste, with several grammar, spelling and punctuation problems (see a few examples in the technical comments). Also, check things such as "(?)" in L402 and 427.*

AR: The authors agree that there was room for improving the narrative of the paper. In the revised version of the manuscript we have re-written the introduction and discussion consequently. The "(?)" resulted from the incorrect citation in LaTeX, these typos and other minor errors in the main text or citaions have been revised and corrected. The english languange has also been imporved through the manuscript.

AC: The introduction and discussion were re-written.

*2. Throughout the entire paper, there are several assertions without references (see comments below). For instance, L18-19 or L22-23 in the introduction.*

AR: The reviewer is right, thank you for this comment. We agree that those sencentes needed a reference.

AC: The refences following references were added for the given specific examples: Levy et al. (2012), Bringing physics to life at the submesoscale, Geophysical Research Letters (14), 1-13. Mahadevan (2014), Ocean science: Eddy effects on biogeochemistry, Nature (7487), 168-169.

*3. In the introduction, the motivation of the study, the relevance of the topic and knowledge gap(s) have to be better introduced and established. For instance, I find interesting studying the effect of (sub)mesoscale processes in coastal areas due to its complexity, and maybe there are few studies about it in the BoB. However, L58: "Few studies provide the link between phytoplankton occurrence and physical processes in the BoB" is definitely not true. In relation to this, maybe it could be highlighted and argued the importance of using data collected at different spatiotemporal scale (some- thing this paper does well). Also, I do not feel that the introduction goes from general to more particular aspects of the research topic. In some paragraphs, there are too many ideas (although there should 1 or 2), which is especially clear in paragraph 1 (L18-29). Why not using a whole paragraph (or even 2 paragraphs) to develop the importance of (sub)mesoscale processes and phytoplankton and their link? Why focusing so fast on eddies when this is just one ase? Why introducing already the BoB? Other paragraphs just feel like a collection of examples with no clear message (for instance L58-67: paragraph 4). Additionally, the ideas and concepts have to be connected better, within and among paragraphs.*

AR: The introduction has been restructured considering the reviewer's remarks, some sentences were carefully re-written, some references were added to better support the presented topic. Indeed, there are some publications dealing with phytoplankton occurrence and physical processes in the BoB (Bode & Fernández, 1992; Fernández et al., 1993; Herbland et al., 1998; Lampert et al., 2002; Labry et al., 2001). However, not many of them have dealt with fine spatial scale distribution (Lunven et al., 2005, Smythe-Wright et al., 2014) and/or with hydrological and hydrodynamic measurements (Zaráuz et al., 2007, Muniz et al., 2019), combined to phytoplankton high resolution spatial distribution resolved at least at the pigmentary/functional level. Sub-mesoscale processes are more and more considered to be essential to determine the spatial variability and also the dynamics of phytoplankton in marine coastal and shelf studies. We consider this aspect should be introduced more clearly and have built a specific paragraph to this end.

AC: New paragraph : The interaction between ocean dynamics and phytoplankton covers a wide range of spatio-temporal scales, and these are inherent to the surveying strategy to be selected. D'Ovidio et al. (2010) linked the occurrence of different phytoplanktongroups with the large scale surface ocean dynamics, based on altimetry data. They defined the so-called fluid dynamical nicheswhere the phytoplankton assemblages interact with distinct physiochemical environments. However, available satellite observations lack the spatio-temporal resolution and/or coverage to properly resolve the fast-evolving (sub)mesoscale coastal processes. In coastal regions whereoceanic currents meet the bathymetry, the connection between the (sub)mesoscale processes and phytoplankton becomes even more challenging and therefore requires more demanding surveying methods to be able to cover a wider range of spatio-temporal scales. Gliders which can typically cover 1 km horizontally in an hour are also too slow for large features of O(10) km. An alternative is ship-towed undulating devices, which allow sampling 10-20 times faster than a glider (Lévy et al., 2012). Contrarily, (sub)mesoscale to microscale vertical patterns of chlorophyll-a (chl-a) concentration have been studied widely bythe use of in vivo fluorometric casts, allowing to identify the Deep Chlorophyll Maximum (DCM) (Cullen, 2015). Differences within the DCM in terms of concentration, biomass and diversity (Latasa et al., 2017) stress the importance of the environmental drivers involved, on which the (sub)mesoscale processes play a critical role (Lévy et al., 2012).

*4. The statistical analyses should be better conducted. This also affects section 3.3 in results. Although I like the use of GAMs, I really miss a model comparison based for example on AIC (or other criteria). Why are predictors inspected only one by one? What about interaction effects among predictors? In GAMs, those could be included as tensor products or varying coefficient models. Have the authors checked if the predictor variables are correlated? I think using a model with interaction is a better approach than the analyses described in L160-162 and L339-348. Also, in material and methods, the description and specifications of the model(s) and statistical methodologyis too scarce. What are the formulas and specifications for the model(s) used? What are the characteristics of the residuals? In results, what about deviance or variance explained by the models?*

AR: The authors agree that there was room for improvement on the description and implementation of the statistical analysis. We explored different combinations of the models by removing variables in order to assess the independent contribution of each of the variables had on the model, and we compared them based on GCV. This gave us information about effects among predictors. The analysis described in L160-162 and L339-348 has been substituted for an additional GAM analysis, which now covers the DCM. We also added a table for each GAM containing the deviance explained by each variable according to the method followed in Llope et al. (2009).

AC: A new section was included in Material and Methods with a specific explanation of the statistical analysis under the section called "2.4 Statistical Analysis". Figures, tables, results and discussion have been updated to include the new analysis.

*5. Sometimes, the study mixes different biological concepts. For instance, fluorescence can be translated to Chl a concentration. However, Chl a concentration is not the same as phytoplankton biomass, as this depends on variations of C:Chl a cell ratios. In addition, larger biomass accumulation does not necessarily imply larger phytoplankton growth. See for instance L263 and L361.*

AR: The data presented as total and spectral group fluorescence are in fact Chl-a Equivalents units concentration (µg ChlaEq L-1) after manufacturer's calibration with microalgal cultures. Therefore, they are not technically raw fluorescence data (the units label were corrected in the MS). We agree that changes in Chl a Equivalents does not necessarily reflect changes in biomass. In addition, we agree that changes in biomass do not imply higher in situ growth as photo-acclimation and physiological status can lead to changes in C/Chl-a ratios, as well as advection/sedimentation/migration processes could concentrate phytoplankton biomass that were produced in other place and time (Durham & Stocker 2012: Wirtz & Smith 2020). We thank the reviewer for this relevant comment. In addition, we also modified the title.

AC: Thus, sentence in Lines 262-263 were changed as follows: "The GAMs shown in Figure 8 correspond to the section "Above the Pycnocline" where low salinity values, and hypothetically high nutrient concentration related to river discharge, exert the greatest impact in the chl-a and explain 18.2% of the total chl-a deviance." In addition, the fluorescence units have been changed to µg ChlaEq L-1 in the entire manuscript.

*6. In my opinion, the authors should translate better the relationships found into a mechanistic understanding. How could salinity affect distribution? Is it directly affecting phytoplankton physiology and growth? Is it because differences in salinity reflect the occurrence of processes such as fronts or other modifications of the physical structure of the water column? The same goes for vorticity.*

AR: We thank the reviewer for these ideas. Indeed, we believe that the variations in salinity are very small to affect the physiology of the phytoplankton. It is more likely that it reflects an effect of nutrient-availability related to river discharge above the pycnocline, whereas below it corresponds to deep water repleted in nutrients. Vorticity on the other hand might reflect the advection to the core of the anticyclone by Ekman transport.

Althoug the present study offers an valuable opportunity to examine physical-biological coupling we are aware of the limitations ot the data set analysed here. For instance, in the lack of nutrients or O2 measurements, a unique interpretation of the results in terms od phisiology and dynamics of phytoplankton is complex. Thus, along with clearly stating the limitations of the study we have tried to improve the discussion on these points in the manuscript.

AC: The following two sentences are added in at different paragraphs in the discussion in relation with the possible mechanisms affectinghe chl-a concentrations: "Overall, when integrating to the entire water column, even though the responses differ in the different sections, salinity is the most important environmental factor regarding the Total chl-a distribution and the relative occurrence of Brown and Green algae. We attribute this effect to salinity and its relation to nutrient content at the surface fresher and at the deeper saltier waters" and "We believe that the observed submesoscale processes during the Etoile campaign would have perturbed an already existing horizontal layer of DCM, not enhancing primary production (not measured during our study) by themselves, but rather isolating, advecting and gathering the phytoplankton in the region of anticyclonic circulation."

*7. In my opinion, the discussion is the weakest part of the article. It has to be better structured and the implications more clearly defined at the end (including the conclusions). Please, be more concrete, thinking about how specifically does this study contribute to the field (the statements at the end of the conclusion sounds too generic).*

AR: In line with what has been argued in the previous comment the discussion hast been thoroughly reviewed and restructured.

AC: A first paragraph summarising the main results was added. In addition, the discussion section has now been divided in the subsections. (1) Physical Environment: which discusses the hydrographical and hydrodynamic features that were present during the Etoile oceanographic cruise; (2) Environmental Drivers: Here we discuss the effect of the different environmental factors (salinity, temperature, vorticity and vertical velocities) on the distribution of chl-a in the different sections of the water column, both from observations and from the GAMs; (3) Limitations of the study: Here we discuss the limitations of the study in terms of variables that were not measured and the possible sources of uncertainty regarding the use of chl-a as a proxy to phytoplankton biomass.

**Specific responses:**

Abstract

*L2-4: Why just focusing on the effect on nutrients (which are not directly analyzed in the study)? What about the modification of the water column structure? They only define niches or also affect phytoplankton through advection? I like the motivation for the study in L4-5.*

AR: We agree that the nutrients are not the central topic in the present study, and therefore we now mention the modification of the water column and active gathering through advection.

AC: The monitoring and characterization of submesoscale dynamics are determinant for the appropriate comprehension of marine ecosystems (Levy et al, 2012). Submesoscale processes refer to those features that range on spatiotemporal timescales of O(0.1 - 10) km and O(1) day. The timescales at which these processes evolve make them uniquely important to the structure and functioning of planktonic ecosystems (Levy et al, 2012, Mahadevan 2016). They interact with the ecosystem by either driving episodic nutrient pulses to the sunlit surface, by increasing the mean time that the photosynthetic organisms remain in the well-lit surface (Levy et al., 2012), or even by reducing and even suppressing the biological production (Gruber et al., 2011).

*L11: This is again the goal of the study. Merge it with L6-7.*

AR: Done.

*L15-16: Do studies analyze hydrographic aspects and not consider the dynamics of the system? Think about more particular and specific implications of this work.*

AR: Although other studies consider the dynamic variables they have not statistically constrained their effect.

AC: The present study tries to statistically constrain the effect of the dynamic variables on phytoplankton distribution.

Introduction

*L18: Please define spatiotemporal scale ranges for mesoscale processes (km and days). For instance, Mahadevan (2016) in L22 actually talks about submesoscale processes. Maybe mesoscale should be replaced by submesoscale in the entire article (or at least in some parts)?*

AR: We have decided to substitute mesoscale by submesoscale in the entire article.

AC: submesoscale dynamics are now defined on a the spatiotemporal scale of 0.1 - 10 km and days.

*L20: Not only nutrients can be limiting. Light is the other main factor (e.g. in temperate and polar areas in winter or at depth). Elaborate.*

AR: We agree that light also can be limiting.

AC: We mention the two ways that submesoscale processes affect the nutrient supply and the light availability: "They interact with the ecosystem by either driving episodic nutrient pulses to the sunlit surface, by increasing the mean time that the photosynthetic organisms remain in the well-lit surface (Levy et al., 2012)".

*L20: Are not all processes confined in time and space? What does this mean?*

AR: With this we referred to the fact that these processes have specific spatiotemporal scales, however, we understand that the sentece is confusing.

AC: This sentence was changed to "submesoscale processes refer to those features that range on spatiotemporal timescales of O(0.1 - 10) km and O(1) day. The timescales at which these processes evolve make them uniquely important to the structure and functioning of planktonic ecosystems (Lévy et al., 2012; Mahadevan, 2016)."

*L21: After "evolve and transport seawater properties" there is a "-i.e. nutrients". However, this is just one of the non-conservative properties.*

AC: This was corrected in the new version in addition to the restructuring of the whole section.

*L31-40: Give a purpose to the paragraph by highlighting that the study area is complex in terms of coastal hydrographic processes.*

AR: Done.

AC: The following sentence has been added: "The BoB is an area of complex coastal hydrographic and hydrodynamic processes, mainly due to the intricate bathymetry, the seasonally modulated and episodically strong river runoff, the wind- and density-driven ocean circulation and their interplay".

*L44: Include a mention to a figure in the introduction is unusual as far as I know.*

AC: The mention of the figure has been removed.

*L76-78 should be integrated with L71-72. Both parts are about what was done in the study.*

AR: Done.

Material and Methods

*L90-102: Include references for all the instrument and methodology used. Also, is there any calibration? Chlorophyll is mentioned here, but fluorescence is used for the analyses instead. Four algal pigmentary groups can be detected by the fluorometer, but only 2 are inspected in this study. Why? If there were an instrumental bias and different measuring sensitivity for the 2nd multi-spectral fluorometer, can the data be trusted at all?*

AR: The Fluoroprobe was calibrated by the manufacturer with a standard procedure both for translating fluorescence into chlorophyll-a equivalents, as well as for differentiating up to 4 micro-algal pigmentary groups (Beutler et al., 2002; MacIntyre et al., 2010 ). No bias was detected from the two machines. The profiler just did not record substantial signal attributed to cyanobacteria ("blue-green algae" nor "red algae"). However, a difference in the groups determined was effective when comparing continuous recording surface waters and surface signal in vertical profiles. An hypothesis explaining the differences could be Non-Photochemical Quenching- NPQ) for in situ profiles at maximum lignt levels from one side, and/or a possible interference with dissolved fluorescence matter in surface continuous recording (even though yellow substances are retrieved from the signal). However, when comparing the sub-surface total chlorophyll estimates to some chlorophyll-a concentrations assessed on filters (data not shown), a good correlation was found. That's why we kept (for the high spatial resolution study of sub-surface waters, at least the total chlorophyll-a estimates of continuous recording system (but did not use the spectral discrimination).

*L113: Why a 3h running mean average was applied? Please explain and justify.*

AR: The 3-h running mean is applied to the radar radial velocities as a pre-processing step to smooth radial fields and ensure more consistent total velocity data.

AC: Then, a centred 3h running mean average was applied to the resulting radial velocity fields as part of the pre-processing previous to the computation of total currents.

*L135: Define small Rossby number range. Is it realistic to assume totally constant mesoscale features during the sampling? If not, elaborate this. Also, replace time/space by spatiotemporal. In relation to this point, is it not always important the spatiotemporal distribution of observations?*

AR: The mesoscale processes occur in a timescale of day. The eddy is identified by the HF radar between the 26 and 29 of July, and again on the 6 to 9 of August. Even if there is a slight southward migration, it shows a persistent nature. The hydrographic data collection last for 3 days, which we believe is fast enough to capture a snapshot of the ocurring processes.

AC: We know define Rossby number in the manuscript as "$Ro = U/f\, L \ll 1$ (where U is the characteristic velocity, L is lenght scale and f the Coriolis paramenter". Time/space was replaced by spatiotemporal.

*L139: Which are those key dynamical variables? Please elaborate.*

AR: Those variables are the geostrophic relative vorticity and vertical velocities

AC: "geostrophic relative vorticity and vertical velocities" was added to the text.

*L148: "the correct representation" is a strong statement. Better "good" or "appropriate".*
*L153: Include the statistical analyses as a different section with a title (2.4).*

AR: Done.

AC: "Correct" was changed by "appropiate"

*L159: Please indicate the version of R and mgcv package used. Also, cite the R Program.*

AR: Done.

AC: R (version 3.63, R Core Team (2020) and mgcv (version 1.8.33).

Results

*L166-168: Part of it belongs to Material and Methods or even to the introduction.*

AR: The authors agree with the reviewer and part of these sentences and the impact of the wind in the circulation is now explained in the introduction.

AC: In the introduction: "The circulation in the coastal SE BoB is controlled mainly by the prevailing winds".

*L171: "relaxed" is in my opinion vague.*

AR: We removed part of the text.

AC: "As mentioned before, wind intensity and direction play a major role in determining the surface oceanographic setting in the SE-BoB" was erased as this information is already given. "Relaxed" was removed.

*L186:  Do these results belong to the 1st or 2nd period? Clarify this.*

AR: The results belong to the 2nd period.

AC: The following line was added: "The sampling was carried after the wind shifted to a north-easterly component.

*L219-220, L222-224, L246-256 and  L266-271 and  L288-293: Belongs to the  discussion.*

AC: These sentences were reformulated and moved to the discussion section.

*L238:   It is hard  for me  to observe how  green algae fluorescence follow the  salinity contours at waters saltier  than  35.49. Please clarify this.*

AR: 35.49 was a typo, the correct contour is 35.55.

AC: 35.49 was changed by 35.55 .

*L239:  "logarithmically  transformed" sounds better to me.*

AR: Done

AC: "Logarithmically normalized" was changed by "logarithmically  transformed".

*L265-266: "isohaline"  instead of "halocline"?*

AR: Yes

AC: Changed

*L266-271 and  L288-293: Check the  writing  and  description of the  relationships (for instance, I do not observe a positive relationship at the edges of the range of temperature).*

AR: This relation is for the total chlorophyll-a as a result of the differential effect on brown and green algae.

AC: This whole section was re-written and restructured.

Discussion

*L260-307: I think the first paragraph should be a brief summary of the main results.*

AR: We agree that this would make the whole discussion section much more clear.

AC: A new paragraph explaining the main result was added at the beginning of the Discussion section "Prior to the Etoile campaign, two cyclones (C17w and C17E) were observed by the HF radar, these disappeared from the surface signal when the wind changed by the time the campaign took place. However, their signal remained at subsurface and could be diagnosed from the hydrographic measurements during Etoile. From the derived geostrophic circulation, a dipole structure was observed, an additional anticyclone (A17) together with a region of anticyclonic circulation between C17W and C17E were recorded. Further two salinity fronts were observed, one at the surface ($<$14 m) and a second deeper one ($>$50 m). From the chl-a observations, a DCM was observed below the pycnocline at $\sim$60 m. By measuring the chl-a of different spectral groups of algae we depicted the two dominant groups, Brown and Green algae. The relative importance of the environmental factors modulating the chl-a distribution was assessed with GAMs. The GAMs showed not only that these environmental factors affect the Brown and Green algae differently, but also that their relative importance changes throughout the water column. While salinity and temperature explain most of the deviance above and below the pycnocline of both Brown and Green chl-a, it is vorticity that captures most of the deviance in the DCM for Brown algae".

*L301-305 belong part to Material and Methods and part to Results.*

AR: Although we agree that part of the listed sentences is part of the methodology "The extension of this low salinity front over 20 km horizontally and 18 m vertically (Figure 5) if we consider the boundary at a salinity of 35.1 (Puillat et al., 2006)", we decided to move everything to Results. This specific section explains the criteria for describing the results once the actual results have been shown and therefore it cannot be introduced before some of the results have been described.

AC: The specified lines were integrated in the Results section.

*L318: What do non-linear terms mean here? Join this paragraph with the previous one?*

AR: Non-linear terms refer to the frontal instabilities from which the submesoscale processes arise (Levy et al., 2012).

AC: Since the term is confusing and the paragraph in general is not applicable to the rest of the study (no biological data was collected in A17), we decided to remove them.

*L327-328: Is this statement from other studies? Then, please include references. If not, from the results provided in this work, this can only be speculated (nutrients were not analyzed here). Riverine plumes have also other effects such as advect phytoplankton or generate fronts were plankton can accumulate.*

AR: Yes, but other studies stated that nutrients are well depleted since late spring in the BoB (Muñiz et al., 2019), especially out of the direct influence of big estuarine plumes which can advect phytoplankton or generate fronts that can be observed in surface waters.

AC: The references of Muñiz et al., (2019) and Borja (2016) have been now included.

*L336-337: Again, this should be more speculative.*

AR: Modified.

AC: Nutrients were removed from the sentence.

*L339: See general comment 6. Include references.*

AC: The following references are included Cullen 2015.

*L340-348: This belongs to Material and methods (and part to Results).*

AR: This paragraph was removed since we substitute this analysis for a GAM which focuses on the DCM.

*L350-365: This belongs to Results.*

AR: This paragraph was removed since we substitute this analysis for a GAM which focuses on the DCM.

*L372: What does mean "areas of vertical velocities"? of maximum velocities? Why should we expect higher concentrations in these areas? Elaborate.*

AR: There was a word missing, this was referred to the upwelling areas or "areas of positive vertical velocities". We expect higher chl-a concentrations since these areas ar bringing nutrients from subsurface to surface.

AC: The sentence has being rephrased to " However, the highest phytoplankton concentration does not coincide with the areas of upwelling (positive vertical velocities)".

*L382-383: I do not get this sentence.*

AC: The whole parapgraph was removed from the section since it was more a general statement and examples and not so relevant for our study.

*L409: Include references. Also, diatoms have different mechanisms to regulate their vertical position. This can be discussed too.*

AR: Diatoms were also present (according to some microscopic observations carried out by the DCM) and can indeed regulate their buoyancy by changing their fatty acid and lipids composition. However, as the majority of the species detected around the DCM were dinoflagellates, we assumed that their ability to perform vertical migration was combined to physical forcing to define the DCM. Both diatoms and dinofalegallates can orientate their selves to maximize both light and nutrient absorption (Besterretxea et al., 2020).

AC: References were included but also parts of the paragraphs were removed.

*L418-430: I miss some references here.*

AR: We agree that some references are missing.

AC: We included the following references Latasa et al., 2017 as an example of clear discrimination of phytoplankton diversity by multiple techniques in DCM, as well as Houliez et al. (2012) about the Fluoroprobe factory fingerprints which determined on mono-specifical cultures or target micro-algae which are not necessarily representative to our shelf and ocean system.

 Figures

*Include in the caption all the information necessary to understand each figure. For instance, in Fig. 1 indicate that the dot corresponds to the buoy, stars correspond to radar antennas and names to rivers (do the same for the other figures). Also, before the acronyms such as MVP, include the complete name.*

AC: The captations were improved and in addition fluorescence was substituted by chl-a equivalent units ($\mu$g ChlaEq L-1).

*-Figs. 1 and 2 can be joined in a single figure.*

AC: Done

*-Fig. 1: Are the eddies shown a permanent part of the general hydrography? Clarify.*

AR: The eddies shown in Fig 1 are mostly seasonal and related to the winter strengthening of the Iberian Poleward Current (IPC).

*-Fig. 2: Replace "T-" by: (T-1, T-3 and T-5).*

AC: Done

*-Fig. 3: Include axis with units.*

AR: This figure is a Progressive Vector Diagram (PVD), it represnets the wind direction and intensity, which is described as the black bar (5m/s). This type of plots are presented as it is (See Figure 2C in Puillat et al (2006) (http://dx.doi.org/10.3989/scimar.2006.70s115).

*-Fig. 4: Change labels of facets by A, B; C, D; E, F. Please, do not use scientific notation for the colour scales in this case. Also, note if the scales are logarithmic for turbidity and Chl a.*

AC: Done

*-Fig. 5: indicate in the caption that scale ranges are different for each depth. Salinity has no units, so delete PSU (apply this to other figures and text too). To what date(s) correspond(s) the maps? Indicate what positive and negative vertical velocities mean. The 43.77◦ N dashed lines are hard to see.*

AR: This plot corresponds to a synoptic shot representative of the conditions during the 2nd to the 4th of August 2017 when the ETOILE oceanographic campaign was carried out.

AC: PSU was deleted. We describe what positive and negative velocities mean. Lines were changed to be more visible.

*-Fig. 6: Are isolines actually isopycnals?*

AR:Yes, for A and B the isolines are isopycnals, However for C and D the contours corresponds to geostrophic velocities.

AC: This was specified in the captation.

*-Fig. 8: To which dates correspond the plots?*

AR: This plot corresponds to a synoptic shot representative of the conditions during the 2nd to the 4th of August 2017 when the ETOILE oceanographic campaign was carried out.

AC: This was specified in the captation.

*-Figs. 9 and 10: Capitón should start with "Relationship between XXX and YYY". Please, do not use the default R output and replace variable names in the x-axis by the name of the variable and the units. Why the y-axis (fluorescence) in the 1st 3 rows can be negative?*

AR: The y-axis indicates the partial additive
effect that the term on the x-axis has on the chl-a, which can be either positive or negative. This is also shown in the GAM by Llope et al. (2009) (https://doi.org/10.4319/lo.2009.54.2.0512).

AC: This was specified in the captation and the variable names in the x-axis were renamed.

*-Fig. A2: Why is the cross-section here 43.70◦ N and not 43.77? Is it because C17W moved?*

AR: This figure just shows another cross-section in a different location. It is equivalent to figure 7.

Technical corrections:

*L33: Insert "of" before "the water runoff".*
AC: Done

*L116: Delete "of the".*
AC: Done

*L149: After "Gomis et al. (2001)" something is missing (in? by? As in?).*

AC: Corrected.

*L180: Replace "generate" with "generated".*

AC: Done

*L182: Replace "; as well as" by a comma.*

AC: Done

*L196:  Erase the 1st "wind".*

AC: Done

*L201:  Include  "at" after  "(A17)". Also, replace "however" by "although".*

AC: Done

*L238:  Replace "is" by "are".*

AC: Done

*L368:  Replace the comma after  "at first" by "as" or similar.*

AC: Done

*L486:  Which number is "XXX"?*

AR: This is a placeholder and refers to the number of publication in Azti.

---

## Author Comment (AC3) · 3 Nov 2020

**Dear reviewer,**

**First, thank you foryour careful review of our manuscript and your remarks. They have been really helpful to improve the overall structure and content of the manuscript and we have addressed them into the new version. Now we have re-written the introduction and discussion section to improve their readibility. We hope that thanks to your suggestions we have managed to improve the manuscript, and that it suits now the standards of Ocean Science. Best regards**

**Best regards,**

**Xabier Davila**

AR = Author's response
AC = Author's changes in the manuscript

**General responses:**

*The manuscript describes mesoscale processes in the shelf of the Southern Bay of Biscay and tries to relate that physical environment with the occurrence and distribution of phytoplankton in the area. The approach presented is very interesting and the manuscript provides a detailed description of a snapshot of the circulation in the SE BoB in August.*

*I have to acknowledge that I am not an expert in ocean circulation, so although I found this part well described and thoughtful, I am not fully capable of reviewing the method- ological details of the description of the mesoscale ocean processes.*

*Since my expertise includes the phytoplankton community of the BoB, my main con- cerns are related to the fact that the aim of the manuscript is to relate the physical environment to the phytoplankton community structure, and I found this connection poorly supported by the data presented.*

AR: We thank the reviewer for this valuable comment. We agree that we need to be less assertive when relating the physical environment to the phytoplankton community structure and rather focus on phytoplankton spectral groups distribution / dynamics based on the information that can actually by extracted by the data we have.

AC: We have changed the title of the paper and reviewed the discussion accordingly.

*First, phytoplankton distribution is presented though accessory pigments fluorescence data, which is variable depending on the proportion of accessory pigments with respect to chlorophyll and depending on the proportion of chlorophyll to phytoplankton carbon. I think these fluorescence data do not represent phytoplankton distribution as straight-forward as the authors claim.*

AR: The data presented correspond to an automated in situ approach of the contribution of different pigmentary groups to total clorophyll-a concentration, estimated by multispectral fluorometry (MacIntyre et al., 2010).

*Also, not all phytoplankton groups are presented in the results, only "green" and "brown" algae, which leaves out all the cyanobacteria, very relevant in the phytoplankton community of the BoB in summer.*

*Regarding writing and composition, the manuscript is a bit difficult to follow, the physi-cal part is better explained (although there are some typos and acronyms not defined, listed below), but the biology part is very confusing, with many concepts not fully explained.*

AR: The data presented as total and spectral group fluorescence are in fact Chl-a Equivalents units concentration (µg ChlaEq L-1) after manufacturer's calibration with microalgal cultures. Therefore, they are not technically raw fluorescence data (the units label was corrected in the MS).

We agree on the fact that relationships between fluorescence and chlorophyll-a estimations from one side, chlorophyll to C (biomass) as well as the accurate discrimination of the different phytoplankton groups, depend on phytoplankton community composition, physiology and light history of cells (Lawrenz et al., 2010; MacIntyre et al., 2010; Catherine et al., 2012; Escoffier et al. 2015; Garrido et al., 2019). Moreover, one of the caveats of this technique is that obtained fingerprints are not stable, but vary between species and physiological conditions. Nevertheless, the signal found is strong and correspond to what other studies has identified as the chlorophyll deep maximum with in vivo total chlorophyll a fluorescence.
We can reasonably hypothesize that during the short period sampled, the changes observed might have corresponded to changes in phytoplankton pigmentary composition as no important changes were recorded in meteorological conditions (which could have influenced water column irradiance and, consequently, physiological state of phytoplankton cells which were always measured during day time). Phytoplankton communities in surface waters might have been affected by hourly changes in irradiance and might have been submitted to Non Photochemical Quenching (NPQ) of the fluorescence signal. The comparison between some surface chlorophyll-a concentrations and some chl-a concentrations around the SCM measured on filters (data not shown) confirmed the difference encountered between surface waters and 30-40-50m-depth.

Besides, cyanobacteria were abundant in surface waters (continuous FCM recording, counts not shown) but not very important in terms of total red fluorescence (chl-a indication) what was confirmed by the very low amount of chlorophyll a attributed to this group (as well as to Cryptophytes) by Fluoroprobe "blue-green" and "red" signal (compared to that of "green" and "brown" algae). Therefore we decided not analyzing their variability as the majority of the total chl-a signal was attributed, , to "Green" and "Brown" algae, according to the Fluoroprobe and manufacturer algorithms (Beutler et al., 2002).

AC: The manuscript was improved to make it easier to follow and concepts are now better explained. Several of these statements are now explicitly included in the main text of the manuscript to clearly show what are the limitations and potential of the data used in this study.

**Specific responses:**

Introduction

    *25 That's an unclear sentence, it is not clear which is the subject (it?) of the first part.*

AR: The authors agree that the sentece is unclear.

AC: This sentece was removed since part of the introduction was rewritten.

    *27 "this cross-self transport" does refer to the complex ocean dynamics mentioned before (26)?*

AR: Yes.

AC: This sentence was also removed due to the restructuring of the introduction.

    *64 Some word is missing here: "different phytoplankton groups" or "different groups of phytoplankton".*

AC: We changed the wording to "different phytoplankton groups".

    *74 MFSD not defined.*

AC: This sentence was also removed due to the restructuring of the introduction.

Material and Methods

*90 Is the FluoroProbe deployed together with the CTD casts?*

AR: Not simultaneously but it was  deployed at the same stations right before the CTD casts.

*95 Typo:  "Cryoyptophytes" should read Cryptophytes.*

AC: Corrected.

*138 I think there is a typo here: "enough resolution for resolving".*

AC: Corrected.

*149  Another typo,  a parenthesis or  a preposition is  missing:  "of the  analysed  field (Gamis et al. 2001)".*

AC: Corrected.

*150 The  treatment of fluorescence data is not very well explained. Only the method to interpolate the values to a regular grid is explained. But regarding the FluoroProbe data themselves, if FluoroProbe provides Chla values (95) why are  they not showed and  it is instead fluorescence? Are the  fluorescence values calibrated with filtered  samples  in any way? Even though chlorophyll is not the same as phytoplankton biomass (given the variability  in the chlorophyll to carbon ratios), it is more  interpretable and  comparable among groups than fluorescence.  Fluorescence is  also  variable depending on  the content of accessory pigments which is also  subjected to photoacclimation and  hence variable with phytoplankton physiological state.  That's for me  the  weakest point  of the  manuscript, that  the  fluorescence values presented hardly  represent the  actual biomass or abundance of the phytoplankton community.*

AR: The FluoroProbe data characteristics and limitations is now more clearly explained and results are now presented in chl-a Equivalent values according to manufacturer's calibration. Some filtered samples were taken to measure chlorophyll-a concentration,  mainly in surface waters and at one deep sample near the DCM. Even though the relationship was significant, we decided to use the manufacturer's calibration to express our results in terms of chl-a equivalents concentration. Presenting the values in chl-a also allow us to make comparable the results among groups. Besides, no significant meteorological changes occurred during the survey, therefore we assume that during the short-term study described in our manuscript not big physiological changes have occurred from one profile to another at the same depth.

We agree that the relation between the actual phyoplankton biomass and the total chl a fluorescence is not straightforward. However, many studies dealing with the DCM and physical constrains deal with total chlorophyll fluorescence as the method allowing to record changes at a fine scale. In our preliminary study, we used a multispectral fluorometer in order to have a first idea of the different pigmentary/spectral groups or signal that contributed the most to total chlorophyll-a fluorescence at

different depths. Unfortunately, we could not get a detailed information of the distribution of phytoplankton taxa and cell abundance. We are conscious of the need, for further studies, to make as much sampling as possible, with horizontal hydrological bottles (as Lunven et al., 2005) to be able to catch the thin layers of accumulation of the different phytoplankton taxa by different complementary methods as microscopy, pigment analysis and flow cytometry (as Latasa et al., 2017) .

AC: We provided a more detailed information: During the cruise, chlorophyll-a (chl-a) was estimated by a FluoroProbe (Bbe Moldakenke) multi-spectral fluorometer, which measures chl-a and accessory pigments using LEDs with different wavebands. Therefore, it is possible to distinguish between four algal pigmentary groups: "Blue algae" (e.g. Cyanobacteria), "Green algae" (e.g.Chrolorophytes, Chrysophytes), "Brown algae" (e.g. Diatoms, Dinoflagellates) and "mixed red group" (e.g. Cyanobacteria, Cryptophytes). It estimates chl-a equivalent concentrations for these four groups and total chl-a following the algorithms of (Beutler et al., 2002) as explained in (MacIntyre, 2010) and a manufacturer's calibration, and also provides an estimation of the concentration of chromophoric dissolved organic matter (CDOM or yellow substances).

*160 This methodology is not very clear, "smaller subsets in relation to the fluorescence" do refer to the spectral groups retrieved by the FluoroProbe?*

AR: This section referred to the filtering technique we applied in the old version of the manuscript.

AC: This sentence has been removed in the new version of the manuscript.

Results

*182 Correct punctuation: "the distribution of the SST, as well as the position of the river plumes".*

AR: Agree.

AC: Changed

*Figure 5 (186) It seems the names of the eddies are duplicated in panels.*

AR: This was fixed in the revised MS.

*Figure 7 (226) Please, indicate which are the units for fluorescence, even if they are arbitrary units.*

AR: The units are in fact Chlorophyll a equivalents (µg chla Eq L-1)

AC: The units were changed.

*Figure 8 (230) Why values of total and groups of phytoplankton are not given in chlorophyll if the output of FluoroProbe is equivalent chlorophyll (95)? Maybe explaining the FluoroProbe technique with more detail would help with the interpretation of the data, or at least including some references about the technique.*

AC: This was corrected in the revised version.

*246 Figure 8 shows depth profile not surface fluorescence, maybe the text should read: "From satellite imagery and continuously recorded surface salinity and fluorescence data (Figure 4 and 7)".*

AR: Figure 8 was a typo.

AC: Changed.

Discussion

*325 I don't think this sentence is correct. Which varies depending on the position in the water column could be which physical driver affects most the occurrence or distribution of phytoplankton, but not the interplay between physics and phytoplankton in general.*

AC: We agree on that, the sentence is modified.

*333 The authors seem to insist throughout the manuscript on the role of salin- ity/ freshwater as one of the main drivers of the distribution of phytoplankton above the picnocline, which is more likely an effect of nutrient-availability (river discharge re- lated). I would suggest the authors to take care of these kind of sentences that relate so directly salinity and phyto distribution.*

AR: We agree that the nutrient availability related to river discharge is the most likely explanation for the relation between phytoplankton and salinity above the pycnocline.

*339-348 This paragraph seems methods to me, not results. Maybe could be useful to have this paragraph in the methods section where the filtering technique is introduced to help explain its relevance (160), which is not very clear (see below).*

AR: The authors agree that this part of the methodology is confusing and therefore it was removed from the manuscript.

AC: This paragraph was removed from the manuscript since the filtering technique was substituted by an additional GAM which focuses on the DCM.

*350 I don't quite understand the point of this filtering technique. If I understood correctly with each iteration only the larger values are selected, and regarding chlorophyll this eventually considers only the large values in the DCM. But, with larger values cor- relation coefficients are also larger, not necessary meaning a higher correlation among data, so I am not sure that correlation coefficients between iterations are comparable. Also, with each iteration sample size, range and probably also variability are smaller which influences the comparability of correlation coefficients among iterations. I would suggest the authors to clarify the relevance of this statistical analysis.*

AR: The goal of this technique was to remove those areas where the phytoplankton concentration is low that add noise to the relations between chl-a and the environmental variables. Due to this low values of chl-a, the GAM that comprehends the whole section below the pycnocline eclipses the relations in the DCM, which are ultimately the most relevant ones.

AC: Now we have substituted this technique by an additional GAM which comprehends the DCM (> 1.5 chl-a eq µg.L-1). This subset still comprehends the 20% of the data below the pycnocline and the relations are significative. This highlights the difference between in modulating mechanisms for the whole section and the DCM.

*353 "The strong negative correlation points suggest that in general brown algae are highly conditioned by the salinity range". Conditioned by the salinity range in which sense?*

AR: The new GAM for the DCM was performed

AC: This new GAM shows that vorticity is the factor that explains most of the deviance for Total chl-a and Brown algae chl-a, whereas salinity explains most of the deviance for Green algae chl-a and the B:G ratio.

*372 Data presented are not of phytoplankton concentration.*

AR: This was corrected in the revised version : estimation of chl-a due to two spectral groups.

*403 The variable fluorescence to chlorophyll ratios could amplify or decrease the signal depending on if the fluorescence comes from accessory photosynthetic pigments (that increase relative to chlorophyll with depth) or from accessory photoprotective pigments (that decrease relative to chlorophyll with depth).*

AR: In the present study, we assume that the sharp deep equivalent chlorophyll maximum addressed by fluorescence is of high magnitude and that even though affected also by physiological changes, it may reflect a peak in chlorophyll-a concentration and, most probably, a peak in phytoplankton

biomass as one can assume that environmental conditions are not very different from those one meters above or below even though we did not measure them.

*409 "The latter (dinoflagellates) can easily regulate their optimum depth by altering their swimming behaviour." Not sure about that, dinoflagellates can swim but not at the spatial scale necessary to change their position in the water column, working against turbulence, mixing and so on. If I am wrong, the authors should include some reference for this statement.*

AR: Some studies address that issue, that dinoflagellates might be more eager to change that much their position but at low temporal rates (see Wirtz & Smith, 2020). However, we agree with the reviewer that these would not explain big amplitude changes in the water column and definitely not working against turbulence/mixing. That's why even this group would be submitted to hydrological and hydrodynamic forcing, as the other phytoplankton groups.

AC: The sentence was removed from the manuscript due to the restructuring of the section.

*427 Possible references for fluorescence to chlorophyll ratios and for fluorescence fingerprint variable within groups and within populations: Estrada, Marrasé and Salat. In vivo fluorescence/chlorophyll a ratio as an ecological indicator in oceanography. Sci. Mar. (1996) 60(1) : 317-325. Kruskopf and Flynn. Chlorophyll content and fluorescence responses cannot be used to gauge reliably phytoplankton biomass, nutrient status or growth rate. New Phytologist (2006) 169: 525–536.*

AR: We thank the reviewer for these references. Some of them were added to the revised version. Indeed, as no biomass estimations were made, we cannot be sure that there was a deep phytoplankton biomass maximum. Nevertheless, the signal found is very strong and correspond to what other studies has identified as the chlorophyll deep maximum (from total in vivo chl-a fluorescence measurements). However, we acknowledge that pigmentary supposed changes recorded could be due to a strict change in phytoplankton composition or to physiological acclimation, nor to changes in biomass.

*429 "In any case, vorticity creates a dynamical niche that plays a major role shaping the phytoplankton community". I find this is a too ambitious sentence, the "shape" of the phytoplankton community is not fully addressed in the manuscript and hence the major role of these vorticity-created niches has not been really evaluated.*

AR: We agree on this observation. Our study was a first attempt on understanding how physical forcing played a role in chlorophyll a total distribution and by spectral groups, as a proxy of pigmentary groups composition).

AC: The link with vorticity is now explained as it follows: Vorticity is the factor that explains most of the deviance in Total chl-a and Brown algae chl-a concentrations. The more negative (positive) the vorticity, the more anticyclonic (cyclonic) is the circulation and the more positive (negative) is the effect on Brown algae chl-a concentrations. Due to Ekman transport, anticyclones have a small component of the velocity that is directed to the core that is able to gather phytoplankton at its core (Mahadevan et al., 2008).

*435 This last paragraph is a mix of many concepts, phytoplankton functional types, biogeochemical models, harmful algae, fisheries. . . I would suggest to reorganize it and focus more clearly on the aims and findings of the manuscript.*

AR: We agree that too many concepts were included.

AC: In the revised version, we point to the specifics aims and findings of the study: We believe that the observed submesoscale processes during the Etoile cruise would have perturbed an already existing horizontal layer of DCM, not necessarily enhancing primary production (not measured during our study) by themselves, but rather isolating, advecting and gathering the phytoplankton in the region of anticyclonic circulation.

**Conclusions**

*447 ". . . joint analysis of remote and operational together with discrete data. . ." is confusing. Maybe repeat data after remote and operational.*

AC: Done

---

## Author Response (AR2)

**#Reviewer 1**

**Suggestions for revision or reasons for rejection**

The manuscript has greatly improved in this new version in terms of readability and structure and the authors have in general correctly addressed most comments and suggestions. However, I think there are still many minor and technical issues that should be corrected. Specific comments and technical corrections:

AR: Dear reviewer, thank you for your thorough and critic review of our manuscript. Your comments have help us to improve the manuscript. We hope that thanks to your suggestions we have managed to improve the manuscript, and that it suits now the standards of Ocean Science. The specific responses to your comments and the related changes are detailed in the following.

Best regards,

**Xabier Davila**

AR = Author's response AC = Author's changes in the manuscript

**Abstract:**

*L2:* Avoid repetition and improve a bit the narrative by better connecting both sentences.

AR: The authors agree that the two sentences are a bit repetitive.

AC: These sentences were merged in the new version of the manuscript: "Submesoscale processes play a determinant role in several ocean processes by transporting momentum, heat, mass and particles. Furthermore, they can define niches where different phytoplankton..."

L4-5: "this effect is" should be "these effects are".

AC: Done

L5: Submesoscale processes coexist with different spatiotemporal scale oceanic processes always. Instead, I think the authors should somehow highlight that in coastal areas, oceanic processes act together with coastal ones, which makes it even a more complex scenario.

AR: The authors agree. Thank you for your comment.

AC: The sentence has been modified to: "However, to evaluate the effect of this variability is not straightforward in coastal areas, where sub mesoscale oceanic processes act together with coastal ones, resulting in a more complex scenario."

L6: What type of dynamic variables? Please specify. Also, delete "the" before "dynamic".

AR: The authors agree that it should be specified.

AC: "dynamic" was changed to "hydrodynamic, such as vorticity". The sentence is now: "The present study brings into consideration the relevance of hydrodynamic variables, such as vorticity, in the study of phytoplankton distribution, from the analysis of in-situ and remote multidisciplinary data."

L9: The link is always there but I think the goal is rather to understand/describe the link between.

AR: The reviewer is right about this observation.

AC: This sentence was changed to: "The main objective of this cruise was to describe the link between the occurrence and distribution of phytoplankton spectral groups and mesoscale to submesoscale ocean processes."

*L14: Replace by the whole name Deep Chlorophyll Maximum (DCM).* AC: Done

L15: Include the acronym and replace General with Generalized (do this replacement in other parts of the paper, for example in L80). AC: Done

*L16: Now DCM can be used instead.* AC: Done

L18: Use a more generic term than deviance: "variability of total...". Also, include at the end of the abstract a sentence or 2 of main conclusions and/or implications of the study.

AR: The authors agree that the main conclusions of the study should have been added. AC: The term "deviance" was substituted by "variability of total". In addition, the main conclusions of the study were added and the last part of the abstract is now: "However, at the DCM, among the measured variables, vorticity is the main modulating environmental factor for phytoplankton distribution and explains 19.30 \% of the variance. Since its distribution within the DCM cannot be statistically explained without the vorticity, this research brings into consideration the relevance of the dynamic variables and multi-spectral chl-a at high spatial resolution. Only by combining both we were able to determine the relative importance of the environmental variables for different spectral phytoplankton groups at the DCM. "

**Introduction:**

*L21: replace "timescales" with just scales.* AC: Done

*L22: I don't know what the O before the parentheses mean. If this is a typo, please correct it. If not, maybe use a more general notation such as the symbol* ~. *Please, fix this for other cases.*

AR: This is a nomenclature used in papers dealing with physical processes to provide an order of magnitude of the spatial scales. See for instance: https://www.nature.com/articles/s41467-019-10149-5. However, since this comment was raised by both reviewers and we expect this paper to have a mixed audience, we decided to use a more extended nomenclature. AC: In order to reach a wider audience and avoid confusion we changed it by "spatio-temporal scales of 0.1 - 10 km and days".

L23: I think Interact (or interaction) is not the best word, in this and next cases throughout the introduction. Interact is a two-way direction, but these processes rather affect or influence the ecosystem (or in next cases the phytoplankton). Please fix this.

AR: The authors agree with this observation.

AC: The word Interact (or interaction) was replaced in the manuscript for a more precise word, for example : "The IPC is, due to the effect of bathymetry, responsible for..." instead of "The IPC is, by the interaction with the bathymetry, responsible for...". Or the subsection in Results section is now called "Exploring bio-physical impacts" rather than "Exploring bio-physical interactions".

*L24: Remove "the" before "photosynthetic".* AC: Done

*L*25-26: Not sure what is the intention of the sentence. What does it mean that "extends beyond primary production?

AR: The authors agree that this sentence is unclear, it was supposed to be a transition between two sentences, as we consider this unnecessary, it was removed. AC:The sentence was removed and "In addition" was added to the beginning of the next sentence.

L27-28: PP does not absorb CO2, but rather the absorption of CO2 occurs during PP.

AR: The authors agree.

AC: The sentence was changed to: primary production drives the absorption atmospheric CO2".

*L*40: *I* don't think "Contrarily" is the best connector here.

AR: We agree that the choice of the connector is not completely accurate. AC: "contrarily" was substituted by "Regarding phytoplankton distribution,"

L43: Move "Latasa et al, 2017" after "drivers involved". Also, (L56) "Caballero et al., 2016" after "plumes".

AC: Done

L55: Add main or largest before "nearby rivers".

AC: Done, "main" was added.

L62: No need to define again what is the DCM; you can put just DCM.

AR: The authors agree. AC: Done

L66: I think the authors should refer directly to the "submesoscale dynamics" (not just especially) and also that the other studies do not analyze hydrographic and hydrocynamic (erase the "s") mechanisms at the same time.

AR: The authors agree with this observation.

AC: The whole sentence was substituted by: "Nevertheless, to our knowledge, none of these studies have focused on the relative importance of submesoscale dynamics, analyzing hydrographic and hydrodynamic forcing mechanisms at the same time."

L69: Replace "its" with "their".

AC: Done

L75: Replace "the consolidation of" with "consolidate".

AC: Done

L78: To avoid redundancy, replace by "...from remote sensing to in-situ measurements".

AC: Done

L80: This last sentence is redundant. Just include in the previous one the new information that is missing (i.e. "on phytoplankton distribution above and below the pycnocline, and at the DCM"), and erase this one.

AR: The authors agree that the sentence is repeating information already mentioned. AC: The sentence was erased and the previous one was changed to: "Secondly, we investigate the link between the observed submesoscale structures and the distribution of the two dominant spectral groups of phytoplankton above and below the pycnocline, and at the DCM, by performing a set of General Additive Models."

**Material and Methods:**

L88: "Undercover" should be uncover, unveil, unravel...

AC: "Undercover" was changed to "unravel".

L96: Replace by light-emitting diodes (LEDs).

AC: Done.

L98: What do you mean with mixed red group? Cyanobacteria are already in the Blue algae group, which I think is correct. Do some cyanobacteria belong to this group? Please specify this. Also, if the "it" in "it estimates" refers to the FluoroProbe, please replace with this.

AR: The sentence was not clear indeed.

AC: We added "phycocyanin-containing Cyanobacteria" to the "Blue algae" spectral group, and "phycoerythrin-containing Cyanobacteria" to the "mixed-red group spectral group. In addition "it" was replaced by "The FluoroProbe".

*Replacements:* (L109) by long-range high-frequency (HF) radar. (L130) by Muller et al. (2009). (L133) by Sea Surface Temperature (SST). (L146) by Gomis et al. (2001).

AC: Done.

L168: Replace by "sections of the water column" or by "layers of the water column".

AC: Done.

L169-171: Suggestion to improve and simplify the sentence: Therefore, the dataset was divided in three different dynamic sections/layers "Above the pycnocline", "Below the pycnocline" and "at the DCM".

AR: The authors agree with the suggestion. AC: The sentence was changed according to the suggestion.

*L173: Erase this sentence as this was already said in the last sentence of the previous section.*

AC: Done.

L180: An error term should be added at the end of the formula (+ epsilon).

AC: The error term was added in the formula and the following sentences were modified to: "Where *a* is an intercept, *z* is the location in the water column (above or below the pycnocline or at the DCM), the *g*s are nonparametric smooth functions describing the effect of environment on chl-a concentrations and *epsilon* is an error term."

L189: Replace "approached" by "approach"

AC: Done.

L190: Is this Wood 2006? Wood 2000? You have this reference incomplete in the bibliography.

AR: This was Wood 2000. There was a bug on the latex code for the bibliography. AC: The reference is now complete.

L192: Replace by (Llope et al. 2009).

AC: Done.

L196: Please keep the same precision for the values (choose 3 or 4 decimal digits).

AR: 4 digits were selected. AC: The precision now is as such: "from 0.0130 to 0.0125".

**Results:**

L206: Please replace "Then" by Thus or Therefore if this is a conclusion from the previous lines. Also, the Etoile cruise occurred 2-4 August 2017, and according to Fig. 2, the direction of the wind was more or less constant during these dates.

AR: The authors agree with this observation. Even if there is a small change in the wind to a Northern component on August 3rd, this is punctual and it can be said that the wind direction was almost constant.

AC: The sentence was changed to: "Therefore, the wind conditions during the whole cruise remained almost constant in direction and low in intensity."

L207: Replace by "fields; the latter allowed us" or similar.

AC: Done.

Replacements: (L209) "give" by gave. (L210) "provide" by provided or provides.

AC: Done.

L216: insert a comma after 2nd. "sharp change" can be replaced by "sharp decrease" to reinforce the message. In L222, insert the 2nd parenthesis after Figure 4.

AC: Done.

*L224: "35.5" is the minimum value so replace < by ~. Or choose a different threshold (<35.56 or <35.57). Apply this idea to the next threshold mentioned (>35.6).*

AR: The authors agree with this observation. AC: "<35.5" was replaced by "~35.5".

*L228: This was already mentioned and can be deleted.* AC: Done.

L255: Why salinity provides a synoptic distribution of phytoplankton? Move this information at the end of next sentence and link it with the existence of a plume. Also, delete the "the" before "the phytoplankton".

AC: We agree that "salinity" is misplaced and was moved to the suggested position. AR: These two sentences were changed to: "Chl-a data collected at surface by the continuous recording system provides a synoptic distribution of phytoplankton during the sampling period (August 2nd - 4th 2017). Figure 6 illustrates how chl-a distribution is spatially dependent on salinity at 3.5 m depth, related to the position of the river plume."

Section 3.3 is a bit difficult to follow as there is too much information and details. I think it should be simplified, highlighting mainly the relationships and details that are most relevant to the story and main messages of the paper (see for instance the last part of the first paragraph in the discussion and section 4.2).

Another issue in this Section 3.3 that was already mentioned in the previous review is that the description of the shape of the relationships is in general confusing and should be better written and explained. Some examples (but check all of them): (L278) lower salinity values can be associated with higher chl a, i.e. they show a negative correlation/relationship; (L284-285): Chl a of brown shows a dome-shape relationship with salinity, with a maximum at around 35.1, so below the effect of increasing salinity is positive and below is negative; (L285) If the relationship is in general negative, the effect of temperature is negative; (L302) Again a case of dome-shape relationship. (L304) Check this one too.

AR: The authors agree that there was too much detailed information in section 3.3. AC: We have removed those details that are not so important for the main story and we also improved the narrative and description of the relationships. We have reviewed thoughtfully the whole section 3 and improved the writing when necessary. We also added a uniform terminology to refer to the three subsections used for the analysis and defined relative to the DCM and pycnocline depths.

L280: Something is missing in "to the explain the".

AR: "to the" was a typo and has been now removed. The sentence was partially rewritten to avoid repetition of words.

AC: The new sentence is now: Salinity and temperature contribute to most of the deviance of the model and explain the 13.10 % and 9.8 % of it, respectively (Table 2)."

L287: Percentage symbol should be after the value in %23.3. Fix this for the other cases.

AR: This was a typo.

AC: It is corrected in the new version of the manuscript.

L300: Include reference to table 1 and maybe figure 9 after "the deviance". Also, in L319, after "the deviance" but in this case Table 1 and maybe figure 10.

AC: Done.

**Discussion:**

L390-394 belong to results (and even the part of the statistical analysis to M&M). Also, L421-428.

AR: We agree that some of those sentences belong to others section and we thank the reviewer for the observation.

AC: Lines 390-394 were removed since this information is already included in the Results and M&M. Lines 421-424 have now become the beginning of paragraph 3 in section 3.3. Lines 424-428 were removed since the information is already in the Results section.

L391: Delete the "And" before "in addition". Also delete "the" in L401 after "explained by" and in L402 after "deviance is".

AC: Done.

L415: I think this could be better written, something similar to "The negative effect of salinity for values higher than 35.1 (figure 8e) is still present below the pycnocline".

AR: The authors agree that the sentence could be improved.

AC: Following the suggestions of the reviewer, the sentence is now: "The negative effect of salinity for values higher than 35.1 (figure 8e) persists below the pycnocline, but the effect is positive at values equal and higher than 35.6, probably due to higher nutrient levels in deeper waters."

L441: Modify as (D'Ovidio et al., 2010).

AC: Done.

L460: Replace "Will help" by something like "would have helped".

AC: Done.

L471: Should it be "direction and speed"?

AR: Yes, thank you for your comment.

AC: The sentence now reads: "The location of the plume depends on the surface currents, which are ultimately conditioned by the speed and direction of the wind."

**Figures:**

General comment also mentioned in the previous review: include in the caption all the information necessary to understand each figure, including explanation for acronyms (the same goes for tables).

Fig1: Delete "these are" in "these are located every". Also replace "dots represents" by "dots represent" and delete the "to" in "to the HF radar". Replace "data, while big white dots to the" by something similar to "data, and big/large with dots mark the". Additionally, replace "square" by "rectangle" and "zoom in area" by "area zoomed in A".

AR: The authors thank the reviewer for this corrections.

AC: The changes were implemented as suggested and, in addition, the sentence " data, while big white dots to the" is now "data, and large white dots mark the".

*Fig3:* Include in the caption a mention of the Cyclonic eddies drawn in the left column (also in Fig. 4). Replace "to periods" by "two periods".

AR: The authors modified the caption in the figure as suggested. AC: "to" was substituted by "two" and sentence about the eddies was added: "The circles drawn in the left column represent the approximate location of the observed cyclonic eddies (C17W and C17E)."

Fig4: Also mentioned in the previous review: Include a note at the end of the caption that the scale range for each variable is different for each depth. This is important as the reader might try to compare the three depths.

AR: The authors agree with these observations and thank the reviewer for pointing it out.

AC: The following sentence was added at the end of the caption: "The scale range for each of the variables is different for each depth".

Fig5: As this cross section goes through A17 and C17E, I think it would nice to draw them as horizontal lines above each column, with its corresponding color and include reference in the caption (also this could be done in Fig. 7). Include also the period (2nd-4th August 2017). Please do also this in Figs. 6 and 7. At the end of the 1s line in the caption, replace by "salinity (A) and temperature (B) with isopycnals (black and white contours, respectively)". Indicate after velocities that solid or dashed black contours correspond to positive or negative velocities.

AR: We thank the reviewer for the suggestion of including the horizontal extension of A17 and C17E , which helps interpreting the figure. The rest of the suggestions were also implemented. AC: Horizontal lines marking the extension of the eddies were added in Fig 5 and 7. The measurement period was included in Fig 5, 6 and 7. The 1s line in the caption in Fig 5 was replace by "salinity (A) and temperature (B) with isopycnals (black and white contours, respectively)" and also it has been indicated that solid (dashed) black contours correspond to positive (negative) velocities.

Fig6: If possible, draw also the eddies. Replace "the black" by "and black ones".

AC: Done

Fig7: after 43.11°N, include reference to Figs 1,4 and 5 (as in Fig. 5).

AC: Done

**Tables:**

Table1: Is Standard Deviation or Standard Error? you wrote SE. Replace "GAMs for" by "GAMs for the water column sections/layers". Replace "the Deep Chlorophyll" by "at the Deep Chlorophyll". Also indicate that dependent variables are the estimated Chl a concentrations for different algae groups and also define what is B:G (also in Table 2). The p-value is missing for DCM, total Chl a. Also, the precision has to be always the same. For instance, if in a case the deviance explained is 57.10, then it has to be in other case 43.00 and not 43 (always 2 decimal digits). Check also this in Table 2. Additionally, deviance explained and R2 report similar information so pick one of them. Why Intercept, SE and deviance explained is separated from the covariate results and R2? If there is no clear reason, merge both groups of rows as this could be confusing.

AR: Thank you for your comment which have help to make the table more comprehensive. SE refer to Standard Error. There was no strong reason to separate the GCV and R2 from the intercept, SE and explained deviance and therefore everything has been merged. AC: We have applied all the suggestions and the precision wass checked in both tables.

Table2: Maybe better replace "Deviance contribution" by "Variance contribution" as in Llope et al. 2009. The R2 is the percentage of variance explained, in this case by the model. However, you also report a percentage of variance accounted by the model. In the table, both seem

to correspond to different things, in particular R2 correspond to the whole model and the % is related to each environmental variable. Please clarify this in the caption for Stepwise Deletion and Delete-one-covariance. Also, indicate in the caption the meaning of the values in bold. Include all the information in the caption, about the water column sections and dependent variables. Finally, replace "vorticity and salinity" by "vorticity or salinity" as the models include only 1 variable.

AR: Deviance was replaced by Variance, also throughout the rest of the text. We also clarified in the caption what is % referring to in each of the cases Stepwise Deletion and Delete-one-covariance. AC: The caption is now: "Variance contribution of the environmental variables to the estimated chl-a concentrations for the different algae groups and Brown:Green (B:G) ratio and the subsets "Above the pycnocline" (APY), "Below the pycnocline" (BPY) and at the Deep Chlorophyll Maximum (DCM). The left columns in each of the sections show these values for the models after stepwise deletion of the variables listed to the left (first vertical velocities and then temperature). The coefficient of determination (R2) and general cross validation score (GCV) and the percentage of variance (%) correspond to the different models. The last two models included only the variable listed (vorticity or salinity). For the right columns in each section, one variable (those listed on the left) was removed at a time while keeping the rest. While R2 and GCV still refer to the whole model, % is individual and corresponds only to the removed variable. Bold numbers point out the main modulating variable -i.e. The one that, individually, explains most of the variance in the model."

**# Reviewer 2**

**Suggestions for revision or reasons for rejection**

The authors have done a fantastic job reviewing this manuscript. It is now easy to follow; the aims and limitations are clearly defined and it describes much better the results obtained and their implications.

Congratulations for the introduction, it is clear and concise. The discussion repeats a bit results for my taste, but overall is a good discussion, it focuses on the insight that can be obtained from the results and raise interesting questions to be further explored. Great conclusions. I just mention below some details, mostly formal, that I think could tidy up a bit more the manuscript.

**Dear reviewer,**

First, thank you for your careful review of our manuscript and your remarks. Thanks to your comments we have improved the discussion section avoiding repetition with the results. We hope that thanks to your suggestions we have managed to improve the manuscript, and that it suits now the standards of Ocean Science.

**Best regards,**

**Xabier Davila**

22 I don't know if it is intentional or not I don't understand this notation for space and time scales "O(0.1 - 10) km and O(1) day". It appears again in lines 39 and 365.

AR: This is a nomenclature used in papers dealing with physical processes to provide an order of magnitude of the spatial scales. See for instance: https://www.nature.com/articles/s41467-019-10149-5. However, since this comment was raised by both reviewers and we expect this paper to have a mixed audience, we decided to use a more extended nomenclature. AC: In order to reach a wider audience and avoid confusion we changed it by "spatio-temporal scales of 0.1 - 10 km and days".

38 "requires more demanding surveying methods that can cover a high range of spatiotemporal scales." I would say that requires methods that can provide high spatio-temporal resolution more than a range of scales.

**AR: The authors agree with this observation.**

AC: The sentence was changed to: "In coastal regions, where oceanic currents meet the bathymetry, the connection between the submesoscale processes and phytoplankton becomes even more challenging, and therefore requires more demanding surveying methods that can provide a high spatio-temporal resolution".

70 "spectral groups" have not been defined yet, so maybe just "phytoplankton groups distribution".

AR: Done

From line 98 onwards it is clear that fluorescence is now chlorophyll, but in the rest of the methods (101,165,172) and results (257) the authors keep using fluorescence and chlorophyll alternatively. To stick to just chlorophyll would be clearer, I think.

AR: Thank you for your comment, we think you are right. AC: The text has been modify accordingly.

134 The comma between "turbidity" and "from" looks out of place.

AR: Thank you for your remark. AC: The comma has been deleted.

287 Check the placement of some "%" signs.

AC: The typos were corrected.

401 In this sentence, "on the other hand" is not opposing any other previous concept. Also, it is used twice in the same sentence. I imagine "on the other hand" would fit in line 409 at the beginning of the "Below the pycnocline" part as opposed to the "Above the pycnocline" one.

AR: Thank you for your remark.

AC: The sentences have been rewritten: "At APY, most of the variance of Total and Brown algae chl-a is explained by salinity, while the environmental variable that explains most of the Green algae chl-a variance is temperature."

---

## Author Response (AR3)

Author's Response to Editors Comments

**Dear editor,**

**Thank you for your careful review of our manuscript and your remarks that helped to improve the manuscript. We have addressed them in the new version and we are pleased that you consider that this study suits the standards of Ocean Science.**

**Best regards,**

**Xabier Davila**

**Throughout the text, there is no such thing as "multi-spectral chl-a", please include word "fluorescence"**

AC: This was modified throughout the entire manuscript.

**L2, Consider not repeating word "processes" and rephrasing "processes play…role … in … processes." Can you simplify e.g. saying " Submesoscale processes have a determinant role in the dynamics of oceans by transporting momentum …"**

AC: We followed the editor's suggestion and the sentence was chenged to: "Submesoscale processes have a determinant role in the dynamics of oceans by transporting momentum, heat, mass and particles.".

**L5, It is not clear what is the "variability" you refer to, and now the sentence is a bit complicated (by copying directly the words from reviewer). Think what your main message is here and write it direct. E.g. "In coastal areas submesoscale oceanic processes act together with coastal ones, and their effect on phytoplankton distribution is not straightforward." (just an example)**

AC: We followed the editor's recommendation and the sentence was modified to: "In coastal areas, however, submesoscale oceanic processes act together with coastal ones, and their effect on phytoplankton distribution is not straightforward.".

**L 12, add term "fluorescence" i.e. "multi-spectral chlorophyll-a (chl-a) fluorescence profiles"**

AC: Done.

**L13, as above, add fluorescence. "in-vivo chl-a fluorescence"**

AC: Done. This was also modified throughout the entire manuscript.

**L19, I suggest saying "Brown and Green algae groups" to be more precise**

AR: Authors agree.

AC: This was modified accordingly.

**L23, instead of "multi-spectral chl-a" use "spectral groups"**
AC: Done.

**L30, Not necessarily only increasing? Change to "affecting"?**
AC: "increasing" was changed to affecting.

**L33, should read "…absorption of atmospheric…"**
AC: Done.

**L36, Why to state "plankton studies" here in the list? Do you mean it affects how we (should) observe plankton i.e. "monitoring strategies" or something else.**
AR: The authors refer indeed to monitoring strategies.
AC: The sentence was changed accordingly.

**L44, is the use of word "bathymetry" really appropriate here, do you mean "sea floor" or something alike.**
AC: "bathymetry" was changed to "sea floor".

**L75, "chl-a" instead of "chlorophyll"**
AC: Done.

**L90, in the response to reviewers versus modified MS, there is some inconsistency, as word "spectral" is taken out (it is included in your response to reviewer 1. I'm sorry to be picky, but the meaning is very different with or without "spectral" and spectral groups are what you studied. I recommend e.g. saying " … distribution of the two dominant spectral groups of phytoplankton, estimated with multispectral chl-a fluorescence technique, …" Alternatively, as the sentence will become quite long, you may introduce the technique very briefly in the beginning of this paragraph.**
AC: We follow the editor's recommendation and the sentence was modified to: "Secondly, we investigate the link between the observed submesoscale structures and the distribution of the two dominant spectral groups of phytoplankton, estimated with multispectral chl-a fluorescence technique, above and below the pycnocline, and at the DCM, by performing a set of General Additive Models.".

**L117, please note if this second fluorometer is also a Fluoroprobe**
AC: Done.

**L135, the list of references is exhaustive. Typically three-four key references will do: original study, a recent/major modification of it, and a key review. Please reduce the amount.**
AC: Done.

**L149, now you introduce another source of chl-a data. It is thereafter not straightforward to understand which data is used where .It may help if you differentiate fluorescence and satellite based chl-a somehow (e.g. by adding some subscript)**
AC: Done. Satellite data now is mentioned as chl-a$^{sat}$.

**L189 vs. L112, Total chl-a vs. total chl-a, thus need to define "Total chl-a" which is used throughout the text.**
AC: "total chl-a" in L112 was changed to "Total chl-a", since they refer to the same variable.

**L192, should read "Generalized"**
AC: Done.

**L211, strictly speaking, "Green chl-a" and " Brown chl-a" are not defined. And why not state "Brown chl-a to Green chl-a ratio (B:G)"**
AR: The authors agree.
AC: The sentence was modified to "In total, 12 GAMs were carried out from the combination of Total chl-a, Green algae chl-a (Green chl-a), Brown algae chl-a (Brown chl-a) and Brown chl-a to Green chl-a ratio (B:G) among the three vertical subsets (Table 1). "

**L212, "total" or "Total"**
AC: "total" was changed to "Total".

**L213, "Green chl-a" instead of green algae chl-a, or at least be consistent**
AC: Done.

**L243 should read "isopycnal 35.1 as in Puillat et al. (2006)."**
AC: Done.

**L277, should read "assess" (!)**
AC: Done.

**L278-9, Brown and Green should be with capital B and G, following your terminology in Mat&Meth.**
AC: Done, this modifications were also applied to the rest of the manuscript.

**L293-4, In Mat&Meth you have chosen terminology "Brown chl-a", not "Brown algae chl-a", be concise, also found in other parts of text, please check**
AC: Done. The changes were applied throughout the manuscript.

**L409 should read "between chl-a and salinity"**
AC: Done.

**L413-6, remove parenthesis "(negative)" and "(positive or cyclonic)" as they decrease readability, while not bringing any added information**
AR: Agree.

AC: Done.

**L431, "While,", remove comma**
AC: Done.

**L432, "chl-a", not "Chl-a"**
AC: Done.

**L614, Indicate that the following parts (1&2) are for Appendix A: Supplementary Material**
AC: Done.

**Figure 1. Please include word "fluorescence", there is no such thing as "multi-spectral chl-a"**
**Should read "Seasonal to mesoscale"**
AC: Done.

**Figure 2. Somehow you need to indicate that this figure illustrates the satellite data. Figure (including the text) should be understandable alone, thus spell out LRC.**
AR: The authors understand that this comment is for Figure 3 rather than for Figure 2.
AC: The first sentence was changed to" Satellite observations for SST (A,B), turbidity (C,D) and chl-a$^{sat}$ (E,F) corresponding to...". LRC is now spelled out in the corrected version.

**Figure 3. Replace the word "context" by "situation","characteristics", or similar**
AC: "context" was replaced by "conditions" in Figure 4.

**Figure 4. replace "Negative (positive) vorticity values represent anticyclonic (cyclonic) circulation" by "Negative vorticity values represent anticyclonic circulation"**
**If you really need to include further information, add " …while positive values represent cyclonic circulation". Correct also in other figures.**
AC: Done.

**Figure 6. Again, it is not immediate if the chl-a is from satellite, or fluorescence. Please include his information (ok, one gets a hint from colorbar title).**
AR: This figure is derived from the surface Fluoroprobe. \
AC: This was indicated in the figure.

**Figure 7. Last sentence should read " The red and blue horizontal lines represent the horizontal extension of A17 and C17E, respectively, that the section crosses." A similar correction needs to be done in several other sentences in figures.**
AC: Done.

**Figure 7 & 8. Should read "logarithmically transformed" not "normalized"**
AC: Done.